# Mapping the Future Afforestation Distribution of China Constrained by National Afforestation Plan and Climate Change

Shuaifeng Song[1,2], Xuezhen Zhang[1,3], Xiaodong Yan[2]

[1]Key Laboratory of Land Surface Pattern and Simulation, Institute of Geographic Sciences and Natural Resources Research, Chinese Academy of Sciences, Beijing 100101, People's Republic of China.
[2]State Key Laboratory of Earth Surface Processes and Resource Ecology, Faculty of Geographical Science, Beijing Normal University, Beijing 100875, People's Republic of China.
[3]University of Chinese Academy of Sciences, Beijing 100049, People's Republic of China.

*Correspondence to*: Xuezhen Zhang (xzzhang@igsnrr.ac.cn) and Xiaodong Yan (yxd@bnu.edu.cn)

**Abstract.** Afforestation has been considered a critical nature-based solution to mitigate global warming. China has announced an ambitious afforestation plan covering an area of $73.78 \times 10^4$ km$^2$ from 2020 to 2050. However, it is unclear where it will be suitable for afforestation under future climate change. Here, we carried out a finer resolution (25 by 25 km) dynamical downscaling of climate change for China using the Weather Research and Forecast (WRF) model nested with bias-corrected MPI–ESM1–2–HR model; then, using the Holdridge life zone model forced by the WRF model output, we mapped the climatological suitability for forest in China. The results showed that the potential forestation domain (PFD) at present (1995–2014) approximated $500.75 \times 10^4$ km$^2$, and it would increase to $518.25 \times 10^4$ km$^2$, by about 3.49 %, to the period of 2041–2060 under the shared socioeconomic pathways (SSP) scenario (SSP2–4.5). Considering the expansion of the future PFD caused by climate change, the afforestation area for each province was allocated into grid cells following the climatological suitability for the forest. The newly afforestation grid cells would be located around and to the east of the Hu Line (a geographical division line stretching from Heihe to Tengchong). Due to afforestation, the land cover would be modified. The conversion from grasslands to deciduous broadleaf forests in northern China took the most area, accounting for 40 % of the newly afforestation area. The grid cell-resolved afforestation dataset kept consistent with the provincial afforestation plan and the future climatological forest suitability. It would be valuable for investigating the impacts of future afforestation on various aspects, including the carbon budget, ecosystem services, water resources, and surface hydroclimate regime.

## 1 Introduction

Afforestation has been considered a reasonable nature-based solution for global warming (Rohatyn et al., 2022; Yu et al., 2022). Afforestation could increase carbon stocks in terrestrial ecosystems by absorbing atmospheric carbon dioxide through its biogeochemical effect (Jayakrishnan and Bala, 2023; Zhu et al., 2019; Gundersen et al., 2021). Meanwhile, afforestation changes the surface energy and mass budgets, as well as the water cycle by modifying the surface albedo and roughness, as well as the partitioning between sensible and latent heat fluxes (Bonan, 2008; Breil et al., 2021; Wang et al., 2023). Specifically,

afforestation causes warming effects through decreasing albedo and cooling effects through intensifying evapotranspiration, which can partly offset or amplify the cooling effects due to taking up carbon from the atmosphere (Arora and Montenegro, 2011). Afforestation not only impacts climate, but also enhances forest ecosystem services such as maintenance and

enhancement of habitat provisioning and species richness (Brockerhoff et al., 2017). In recent decades, China has implemented large-scale afforestation programs (Zhang et al., 2000), such as the Three-North Shelter Forest Program (Hu et al., 2021), the Grain for Green Program (Xiao, 2014), and the Natural Forest Conservation Program (Huang et al., 2019). These ecological engineering program programs have been beneficial for water conservation (Liu et al., 2023), mitigating climate warming (Yu et al., 2020), increasing terrestrial carbon sequestration (Shi and Han, 2014), reducing water erosion risk (Wang et al., 2021),

and alleviating dust storm (Tan and Li, 2015). These initiatives have significantly increased China's total forest cover from 8.6 % in 1949 to 24.02 % in 2022 (Zhang and Song, 2006; Fu et al., 2023; Moore et al., 2016). It contributed to 42% of the land greening in China during 2000-2017 (Chen et al., 2019).

In September 2020, the Chinese government declared a specific objective of achieving carbon neutrality before 2060 (Liu et al., 2022; Zhao et al., 2022). In pursuit of this goal, China is committed to expanding its forest area in the future, and new

national afforestation plans have been introduced. For instance, the Action Plan for Carbon Dioxide Peaking Before 2030 (State Council of China, 2021) outlines the China's target to increase forest cover to 25 % by 2030. The National Forest Management Planning (2016–2050), issued by the State Forestry Administration of China in 2016, set the afforestation target of about $73.78 \times 10^4$ km$^2$ from 2020 to 2050 in China. Such extensive afforestation in the future would lead to the land cover conversions from non-forestland to forestland, potentially causing series of effects through the above-mentioned physical

progresses. It is crucial that the effects of afforestation are highly dependent on the afforestation location. For example, tropical afforestation may yield greater cooling effects than boreal afforestation (Arora and Montenegro, 2011). However, recent studies find that the afforestation benefits may be overestimated, sometimes is controversial, because the responses of the global carbon cycle to anthropogenic land-use change are uncertain (Bastin et al., 2019; Veldman et al., 2019; Lewis et al., 2019). There is limited climate change net mitigation potential if tree planting in water-limited locations, such as in drylands

(Rohatyn et al., 2022). It is thereby imperative to strategically allocate the national planned afforestation area to specific areas and project the possible land cover changes resulting from afforestation.

Existing researches have studied the climatic effects of future afforestation scenarios (Abiodun et al., 2013; Naik and Abiodun, 2016; Diasso and Abiodun, 2018; Odoulami et al., 2019; Zhang et al., 2022). For example, Odoulami et al. (2019) fully replaced the savanna areas (between 8°N and 12°N) with evergreen broadleaf trees over West Africa to study the climate

effects of future afforestation. An obvious increase in the total annual precipitation was found over the afforested area. Similarly, Abiodun et al. (2013) employed random afforestation scenarios to replace 25 %–100 % of the current land cover in Nigeria and found a local cooling effect. In summary, these existing studies mostly employed idealistic and hypothetical afforestation scenarios, and neglected the future climatological suitability of forest. In addition, process-based dynamic global vegetation models (DGVMs), such as LPJ-GUESS, have been extensively used to explore the responses of potential natural

vegetation distribution to climate change (Hickler et al., 2012; Verbruggen et al., 2021) and are also useful tools to quantify

future afforestation scenarios (Krinner et al., 2005; Horvath et al., 2021). The DGVMs driven by meteorological data generally consider complex biogeophysical, biogeochemical, and physiological progress, such as evapotranspiration, carbon–nitrogen interactions, photosynthesis, and so on (Cramer et al., 2001). Given that the mathematics representations of these processes and their parameters as well as future meteorological scenario data from global climate model (GCM) have large uncertainties, their overlap may yield greater uncertainties (Jiang et al., 2012; Martens et al., 2021).

The impact of future climate change is the most challenging. Previous studies (de Lima et al., 2022; Hinze et al., 2023) explored the responses of potential vegetation distribution to future climate change based on climate-vegetation models forced by the climate projection data of the GCM. However, the resolution of the raw GCM is too coarser (~100 km–300 km) to describe the finer land surface features at a regional scale (Varney, 2022; Turner et al., 2023; Song and Yan, 2022; Parsons, 2020). To overcome such shortage, downscaling techniques are widely used to translate GCM output to regional high-resolution data. Statistical downscaling involves the establishment of statistical relationships between local climate variables and coarsely resolved atmospheric fields (Wilby and Dawson, 2013). However, it is not clear whether this historical statistical relationship is always stable in future climate scenarios. Meanwhile, statistical downscaling cannot ensure the physical consistency among meteorological variables. In contrast, the physically-based dynamical downscaling using a regional climate model (RCM) nested within a GCM could provide high-resolution climate simulations (Giorgi and Mearns, 1999; Mishra et al., 2014). The physical consistency is crucial to identify potential afforestation regions due to the multiple meteorological variables involved. Previous studies (Liu et al., 2020a; Bowden et al., 2021) have employed the dynamical downscaling approach to quantify the climatological suitability of nature vegetation. However, previous studies (Niu et al., 2019; Wu and Gao, 2020) used the raw GCM outputs as the lateral boundary conditions (LBCs) of RCM. It is well known that raw GCM outputs have some uncertainties, and the accuracy of LBCs is the most critical factor affecting the performance of dynamical downscaling due to the underlying biases propagated into RCM through the LBCs (Sato et al., 2007; Moalafhi et al., 2017; Karypidou et al., 2023). Therefore, high-accuracy LBCs are the key to obtaining robust future potential vegetation types. Correcting the GCM outputs before dynamical downscaling is necessary to reduce the underlying uncertainty.

By taking into above mentioned background, this study aims to map the future afforestation distribution in China. It is highlighted that the results are constrained by both the national afforestation plan and future climate change. The national afforestation plan determines the total afforestation area of each province, and climate change determines where it is suitable for forest growth. The introduction is the first section of this paper. The second section will introduce the methodology. The discussion and conclusions are summarized in sections four and five.

## 2 Method

### 2.1 Data sources

This study used three categories of data: (1) ground meteorology measurements data and satellite-observed land use/cover data, (2) national planned afforestation area data, (3) climate modelling data from GCM, and ERA5 reanalysis data.

### 2.1.1 Ground meteorology measurements data and land use/cover data

This study used observed 2 m air temperature and precipitation data from the CN05.1 dataset (Wu and Gao, 2013). This dataset has a spatial resolution of 0.25°×0.25° and a temporal resolution of days from 1995 to 2014. The dataset was produced by interpolating more than 2400 meteorological stations in China using the 'anomaly approach'. The CN05.1 dataset was widely been used to apply to evaluate the performance of regional climate model simulations in China (Yu et al., 2015; Huang and Gao, 2018; Yan et al., 2019; Gao et al., 2023).

The land use type is a key parameter of RCM (Mallard and Spero, 2019; Yan et al., 2021). This study used the Moderate Resolution Imaging Spectroradiometer (MODIS) land cover type dataset (MCD12Q1) for the year 2020 (Fig. S1), with a spatial resolution of 500 m (Friedl et al., 2010). The MCD12Q1 features a 17–class International Geosphere-Biosphere Programme (IGBP) classification scheme (Loveland et al., 2000). It could match the default first 17 categories of land use with the Weather Research and Forecast (WRF) model (Table S1). The MCD12Q1 is highly accurate globally, with an overall accuracy of approximately 75% (Friedl et al., 2010; Sulla-Menashe et al., 2019). It was widely used to investigate land use and land cover change (You et al., 2020; Hou et al., 2022) and served as lower boundary conditions for climate modelling (Yu et al., 2017; Ge et al., 2020; Zhao et al., 2021).

### 2.1.2 National planned afforestation area data

This study also used the national planned afforestation area data, which was from the National Forest Management Planning (2016–2050) (NFMP) released by the State Forestry Administration of China (2016). The NFMP presented the total national afforestation area of $73.78×10^4$ km$^2$ (equivalent to an increase in China's forest cover by 7.7%) and the area corresponding to each province between 2020 and 2050 (Fig. 6d). The NFMP was utilized as a policy constraint to identify the future afforestation domain in China.

### 2.1.3 Climate modelling data and ERA5 reanalysis data

To select the optimal LBCs from GCM, Song et al. (2023) comprehensively evaluated the performances of GCM involved in the Coupled Model Intercomparison Project 6 (CMIP6). It was reported that the MPI–ESM1–2–HR model from the Max Planck Institute outperforms all other GCMs in East Asia. In detail, by comparing with other CMIP6 models, the MPI–ESM1–2–HR model could also represent higher skill in simulating various climatic variables such as the sea surface temperature (Bhattacharya et al., 2022), mean temperature (Karim et al., 2020), total precipitation (Kamruzzaman et al., 2022), large–scale circulation (Han et al., 2022), and so on. The main configuration of the MPI–ESM1–2–HR model utilized in this study comprised the coupling atmospheric (ECHAM6.3) and ocean model (MPIOM version 1.6.2), JSBACH land surface scheme and HAMOCC ocean biogeochemistry model with the spatial resolution of 0.9375°×0.9375° latitude-longitude grid and more model detailed is described in Müller et al. (2018). Actually, the MPI–ESM–MR model involved in CMIP5, which was the precursor of the current MPI–ESM1–2–HR model, had been widely used as the LBCs to force the RCMs to carry out finer-

resolution climate simulation (Kebe et al., 2017; Ozturk et al., 2018; Crespo et al., 2023). It is well known that there are several shared socioeconomic pathways (SSP) for future climate projections in the CMIP6. Here, we used the climate projections of the MPI–ESM1–2–HR model under the middle-of-the-road development (i.e., SSP2–4.5 scenario), which represented the most likely development path to occur (O'Neill et al., 2016).

The ERA5 reanalysis data is the fifth generation global reanalysis product developed by the European Centre for Medium-range Weather Forecast (ECMWF) (Hersbach et al., 2020). The state-of-the-art reanalysis data assimilated multi-source data including ground-based meteorological measurements data, satellite-observed data, and atmospheric sounding data based on a 4D-var ensemble data assimilation system (Hersbach et al., 2020). The 6–hourly ERA5 reanalysis data with a spatial resolution of 1.0°×1.0° from 1994 to 2014 was also used as the LBCs. Climate variables for ERA5 reanalysis data and the MPI–ESM1–2–HR model include atmospheric fields (air temperature, specific humidity, zonal wind, meridional wind, geopotential height) and surface fields (i.e., sea-surface temperature, surface pressure, soil temperature and moisture).

Despite performing better than other GCMs, the MPI–ESM1–2–HR model still exhibits biases. Hence, the corrections of climate mean and variance were carried out using the method referred by Xu and Yang (2012) according to Eq. (1) and Eq. (2). The ERA5 data was used as a reference to correct the MPI–ESM1–2–HR model outputs. The MPI–ESM1–2–HR model outputs were interpolated into grid cells of 1.0°×1.0° using the bilinear interpolation method to match the ERA5 grid cells. The bias-corrected 6–hourly data of MPI–ESM1–2–HR model kept the same means and variances as the ERA5 data (Fig. S2-S3). This bias-corrected approach was applied to the atmospheric and surface fields.

$$H_{cor} = D_{GCM\_H} \times \frac{SD_{ERA}}{SD_{GCM}} + M_{ERA} \quad (1)$$

$$F_{cor} = D_{GCM\_F} \times \frac{SD_{ERA}}{SD_{GCM}} + M_{ERA} + \left(M_{GCM\_F} - M_{GCM\_H}\right) \quad (2)$$

Where, $H_{cor}$ and $F_{cor}$ are bias-corrected data of 6–hourly MPI–ESM1–2–HR models over the historical period (1994–2014) and future period (2040–2060), respectively. $D_{GCM\_H}$ and $D_{GCM\_F}$ indicate anomaly by referring to the historical and future mean of MPI–ESM1–2–HR modeling, respectively. $SD_{ERA}$ and $SD_{GCM}$ indicate the standard deviation of ERA5 and MPI–ESM1–2–HR simulations during the historical period, respectively. $SD_{ERA}/SD_{GCM}$ denotes variance-adjusted term. $M_{ERA}$ denotes the climatological mean of ERA5 data during the historical period and $M_{GCM\_F} - M_{GCM\_H}$ indicates the mean future climate change projected by MPI–ESM1–2–HR.

**2.2 Methodology**

The whole study consists of three steps. As shown by Fig. 1, the first step is to carry out dynamical downscaling and prepare a finer-resolution climate data; the second step is to run the Holdridge life zone model to identify forest suitable lands under future climate change scenarios; finally, the third step is to allocate the national afforestation plan area into grid cells at the size of 25 km, by taking into climatology suitability for the forest.

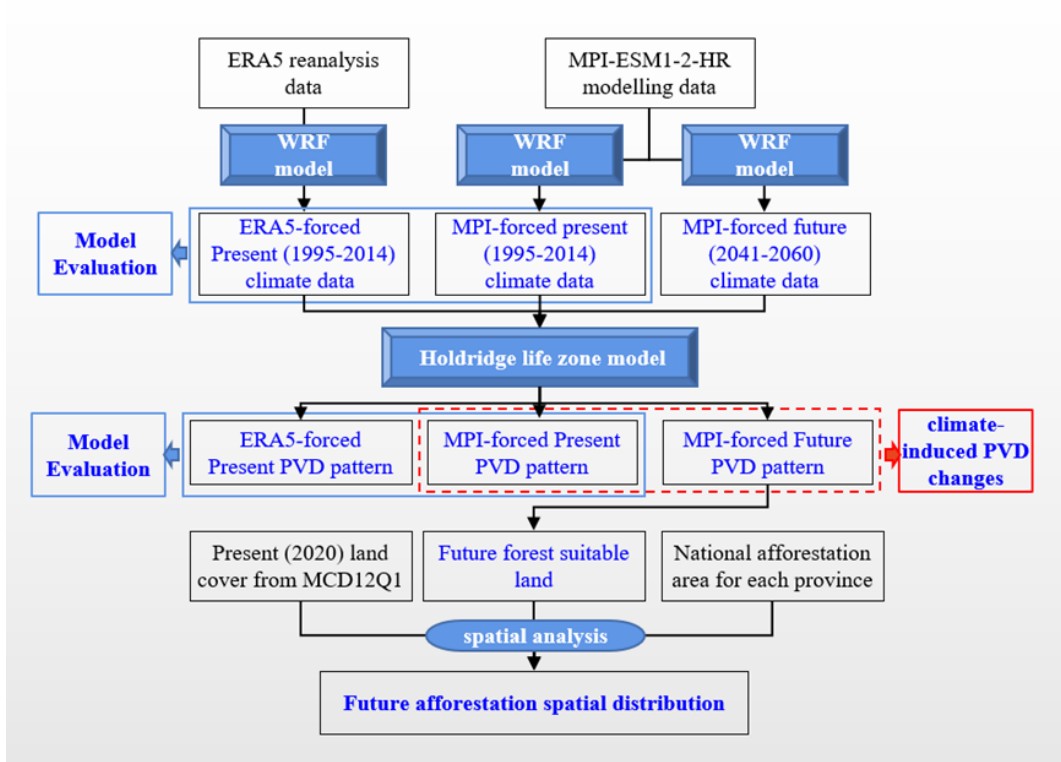

**Figure 1:** Outline for mapping the future afforestation spatial distribution of China (PVD: potential vegetation domain; WRF: Weather Research and Forecast model; MPI: MPI–ESM1–2–HR model)

### 2.2.1 Dynamical downscaling of GCM outputs

In this study, the WRF model served as RCM and was utilized to obtain high-resolution simulations (Skamarock et al., 2019).

As an open-source community mesoscale numerical model, the WRF model has generally been used to investigate regional climate modelling (Wang and Kotamarthi, 2015; Cardoso et al., 2019; Moustakis et al., 2021), whether diagnosis (Ullah and Shouting, 2013; Lu et al., 2021), numerical weather prediction (Case et al., 2008; Zheng et al., 2016), land-atmosphere interactions (Wang et al., 2013; Zhang et al., 2020, 2021). Specifically, the WRF model has been demonstrated to reproduce the historical spatiotemporal characteristics of temperature (Politi et al., 2021), precipitation (Moustakis et al., 2022), and

biomes classified (Zevallos and Lavado-Casimiro, 2022) well, and can successfully project the changes in temperature and precipitation over China (Hui et al., 2018). In this study, the WRF model configurations and physics parameterization (Hu et al., 2015) are detailed in Table 1. The simulation domain is shown in Fig. 2.

**Table 1:** Model configurations and physics parameterization for WRF simulations

| Simulation configuration | Setting |
| --- | --- |
| Model version | WRF version 4.2 |
| Domain | East Asia including the entire China (Fig. 2) |
| Horizontal resolution | 25km |
| Number of grids | 289 (east-west) ×212 (south-north) |
| Vertical layers | 40 |
| Model top pressure | 50 hPa |
| Initial and lateral boundary conditions | ERA5 reanalysis and MPI–ESM1–2–HR |
| Physics parameterization | Optional |
| Microphysics | WSM 3–class simple ice (Hong et al., 2004) |
| Longwave radiation | CAM (Collins et al., 2004) |
| Shortwave radiation | CAM (Collins et al., 2004) |
| Land surface model | Noah-MP (Niu et al., 2011) |
| Cumulus | Grell-Devenyi (Grell and Dévényi, 2002) |
| Boundary layer | YSU (Noh et al., 2003) |

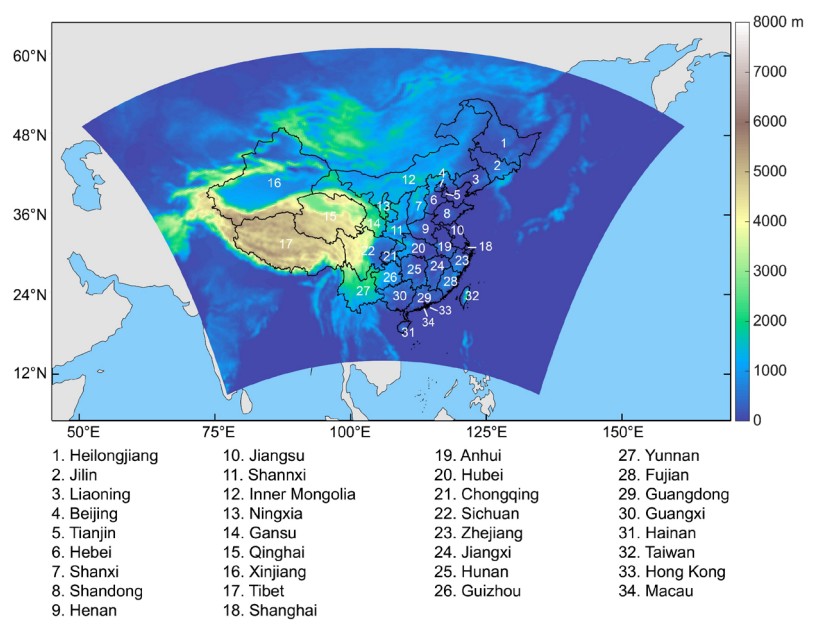

| | | | |
| --- | --- | --- | --- |
| 1. Heilongjiang | 10. Jiangsu | 19. Anhui | 27. Yunnan |
| 2. Jilin | 11. Shannxi | 20. Hubei | 28. Fujian |
| 3. Liaoning | 12. Inner Mongolia | 21. Chongqing | 29. Guangdong |
| 4. Beijing | 13. Ningxia | 22. Sichuan | 30. Guangxi |
| 5. Tianjin | 14. Gansu | 23. Zhejiang | 31. Hainan |
| 6. Hebei | 15. Qinghai | 24. Jiangxi | 32. Taiwan |
| 7. Shanxi | 16. Xinjiang | 25. Hunan | 33. Hong Kong |
| 8. Shandong | 17. Tibet | 26. Guizhou | 34. Macau |
| 9. Henan | 18. Shanghai | | |

**Figure 2:** Model domain with topography. The black boundaries indicate each province in China.

For the historical period, the last two decades (from 1994 to 2014) were considered the historical period in this study because the historical simulation for GCM is up to 2014. Given that the NFMP is implemented for afforestation up to 2050, the simulation for the future period covers the decade around 2050, from 2040 to 2060. Three 21–year numerical experiments were performed using the WRF model (Table 2). The first two experiments, HIS_ERA and HIS_MPI, simulated the historical climate change (1994–2014) using ERA5 analysis and MPI–ESM1–2–HR models as LBCs and default land use, respectively. The future climate change experiment (FUT_MPI) used the 2020 MCD12Q1 land cover in simulating the future period (2040–2060). All the WRF experiments were run for 21 years (1994–2014 and 2040–2060), but the first year (1994 and 2040) as spin-up time was discarded. The remaining 20–year period (1995–2014 and 2041–2060) was analysed. We compared the HIS_MPI and HIS_ERA experiments to validate simulation performance. The FUT_MPI experiment generated a high-resolution future climate dataset under the SSP2–4.5 scenarios.

**Table 2:** Detailed WRF numerical experiment design

| Experiment name | Simulated periods | Lateral boundary conditions | Land use and land cover |
| --- | --- | --- | --- |
| HIS_ERA | 1994–2014 | ERA5 analysis | Default |
| HIS_MPI | 1994–2014 | MPI–ESM1–2–HR | Default |
| FUT_MPI | 2040–2060 | MPI–ESM1–2–HR | 2020 MCD12Q1 |

### 2.2.2 Identify forest suitable lands under the future change scenario

The distribution of terrestrial ecosystems is directly affected by some main climate factors (i.e., temperature) (Piao et al., 2011; Tatli and Dalfes, 2016). Therefore, the impact of future climate change on the forest suitable lands is a further need to be analysed. It is noted that the forest suitable lands in this study indicate the area of the potential forestation domain (PFD). The climate-vegetation models can describe the relationship between the potential vegetation domain (PVD) and the climatic conditions (Dan et al., 2005; Kummu et al., 2021; Anwar and Diallo, 2022). Among a series of climate-vegetation models, such as the Holdridge life zone (HLZ) model (Holdridge, 1947), BIOME4 model (Kaplan, 2001), BOX model (Box, 1981), LPJ-DVGM model (Sitch et al., 2003), MAPSS model (Neilson et al., 1992), IBIS model (Foley et al., 1996), HLZ model is a classification model for simulating the correlation between the potential terrestrial ecosystem types and climate change based on the conjunctions of key climate variables (Holdridge, 1947). In recent years, the HLZ model has been globally well-accepted and used to quantitatively identify the impacts of climate change on the distribution of PVD at the global (Elsen et al., 2022; Navarro et al., 2022), continental (Fan et al., 2019) and regional scales like China (Fan and Bai, 2021; Li et al., 2022). Therefore, the HLZ model was considered to obtain the spatial pattern of forest suitable lands in 2041–2060 under the SSP2–4.5 scenario over China.

The HLZ classification system requires daily temperature and monthly precipitation to obtain three bioclimatic variables: annual average biotemperature (AT), annual total precipitation (TP), and potential evapotranspiration ratio (PE). The output of the FUT_2020 experiment provides these meteorological variables. The HLZ model is estimated with the specific calculation formula as follows:

$$AT(t) = \frac{\sum_{j=1}^{n} T(j,t)}{n}, \quad (3)$$

$$TP(t) = \sum_{j=1}^{n} P(j,t), \quad (4)$$

$$PE(t) = \frac{58.93 AT(t)}{TP(t)}, \quad (5)$$

$$HLZ(t) = \sqrt{(TEM(t) - T_{i0})^2 + (PER(t) - P_{i0})^2 + (PET(t) - E_{i0})^2}, (6)$$

Where, $AT(t)$, $TP(t)$, and $PE(t)$ are the AT (℃), TP (mm), and PE for each grid in the period t, respectively. $T(j,t)$ and $P(j,t)$ are the mean temperature with values above 0 ℃ and below 30 ℃ and the total precipitation on the $j$ th day in the period $t$, respectively. $n$ is the number of days in a year. $TEM(t) = lnAT(t)$, $PER(t) = lnTP(t)$, $PET(t) = lnPE(t)$; $T_{i0}$, $P_{i0}$, and $E_{i0}$ are the reference values of the classification scheme of the AT logarithm, TP logarithm, and PE logarithm, respectively, at the central point of the $i$ th potential vegetation types in the HLZ model classification scheme. $HLZ(t)$ is the $i$ th potential vegetation types in the period $t$. A low HLZ value indicates greater potential vegetation opportunity. Fan et al. (2019) improved the HLZ model and revised the classification scheme applied to Eurasia well. In this study, the reference values of the classification scheme were used to quantify the distribution of potential vegetation types in China (Table S3), and more detail referred to Fan et al. (2019). Compared to the actual vegetation types, the HLZ model can reproduce the potential forest distribution and grassland-forest geographical boundary well (Fig. S4).

### 2.2.3 The approach of the newly afforestation allocation

In this section, we designed an approach to allocate the newly afforestation area for each province into grid cells. To obtain plausible afforestation scenarios, the overall principles were that future afforestation areas should consider both future climate change and national afforestation plan. The specific details are as follows:

(1) The final total afforestation area should be consistent with the NFMP.

(2) Present forestland, cropland, urban, wetland, and water bodies areas do not be encroached on. If the demand of the NFMP cannot be met, we just consider minimizing encroachment on cropland. It can establish the concept of sustainable development as well as avoid repeated afforestation in the future (Zomer et al., 2008). The present land cover dataset for the year 2020 is based on MCD12Q1.

(3) After afforestation, China's cultivated land area is not expected to fall below $121.67 \times 10^4 \, \text{km}^2$ according to the requirements of the National Land Planning Outline (2016–2030) (State Council of China, 2017). This ensures that the cultivated land area stays within the 'red line' and enhances people's welfare.

(4) Afforestation is implemented in areas where the potential vegetation types are forestlands in the context of future climate, according to the output of the HLZ model. This measure could ensure that future climate conditions are suitable for the growth of forests.

(5) Areas with a low HLZ value are allowed priority afforestation. The HLZ metric is a comprehensive metric considering the biotemperature, precipitation, and potential evapotranspiration ratio. A low HLZ value means a greater opportunity to be potential forestlands according to the HLZ model.

## 3 Results

### 3.1 Model evaluation

We evaluate the performance of the key climate variables and PVD based on the observation and two WRF simulations (HIS_ERA and HIS_MPI). The performance of the WRF simulation is quantified by the bias, mean absolute error (MAE), and spatial correlation coefficient (R) for the bioclimatic variables (AT, TP, and PE). Larger R values and smaller bias and MAE values indicate better performance. Figure 3 illustrates the spatial patterns of the observed and simulated multi-annual averaged (1995–2014) AT, TP, PE, and HLZ types in China. The WRF simulation can reproduce well the spatial distribution of the observations with an increasing northwest to southeast temperature and precipitation gradient. However, the underlying bias still remains against the observations (Fig. S5-S7). A more detailed inspection of the scatterplots finds that the spatial correlation coefficient between the observation and simulations (HIS_ERA) is 0.982 for AT, 0.795 for TP, and 0.754 for PE, respectively (Fig. 4). The simulated AT is generally underestimated in most regions, with the national-average bias of -0.974 °C. Consistent with the previous studies (Meng et al., 2018), the largest cold biases are in the Tibetan Plateau and complex terrain region (Fig. S5), with a bias of more than -3.6 °C, which could be attributed to the poor simulation of snow-ice albedo feedback progress (Ji and Kang, 2013). The simulated AT is relatively better in eastern China. The WRF simulation generally overestimates TP in most regions with a national-average bias of 92.883 mm (Fig. 4d). The wet bias could be attributed to inappropriate parameterization schemes (Ou et al., 2020; Zhao et al., 2023), coarse horizontal resolution (Lin et al., 2018; Rahimi et al., 2019), and inappropriate land-surface processes associated with soil moisture and frozen–thawing (Fu et al., 2020; Yang et al., 2018). However, the scatterplot dispersion displays that the simulated TP exceeding 1200 mm in southern China is underestimated (Fig. S5). It is not surprising that the temperature is well-modelled, but the simulation capacity of precipitation-related variables is modest for the WRF model (Gao, 2020). It should be noted that the HIS_ERA simulation exhibits a highly consistent representation to that of HIS_MPI. The cross-correlations for three climate variables between HIS_ERA and HIS_MPI simulation show a high spatial correlation coefficient, and the scatter distribution is very close to the 1:1 line (Fig. 4).

The observed and simulated results of PVD are shown in Fig. 3j–3l. The Kappa statistic is applied to validate the observed and simulated accuracy of the PVD map from the HLZ model (Cohen, 1960). The Kappa coefficient ranges from 0 to 1.0, and the degree of agreement differs across these ranges. According to the description of Landis and Koch (1977), the Kappa coefficient values range of 0–0.2 is considered slight agreement, 0.21–0.40 as fair agreement, 0.41–0.60 as moderate agreement, 0.61–0.80 as substantial agreement, and 0.81–1.00 as almost perfect agreement. Overall, the WRF_ERA simulation could reproduce the distribution of PVD well in China. However, some minor differences in vegetation types are found. For example, in the northeast region of China, the WRF simulation could not precisely reproduce the observed extent of steppe types. Such misclassified zones could be attributed that the model overestimated the precipitation exceeding 220 mm in the transition zone of dry-wet climate (Fig. S5d); thus, the vegetation types are changed from steppe to cool temperature forest. Other disagreement types are found in southern China, where the observed subtropical forest expands northward up to 32°N. However, the simulation results reduce the extent. The dry bias of precipitation simulation in southern China could explain the source of uncertainty. Although the limited ability displayed by the models, the overall accuracy based on the Kappa coefficient indicates a substantial agreement between the observation and the WRF simulation. When compared with the observations, the Kappa coefficient is 0.648 for the HIS_ERA and 0.662 for the HIS_MPI. It suggests that a highly perfect agreement (Kappa coefficient = 0.962) of the PVD between the HIS_MPI and HIS_ERA is shown in Fig. 3k–3l). The implementation suggests that the bias-corrected MPI–ESM1–2–HR model can replace the ERA5 reanalysis data as the LBCs of WRF to obtain similar accuracy in high-resolution simulations. Therefore, in the following analysis, the WRF model forced by the bias-corrected MPI–ESM1–2–HR model will be applied to project future climate change.

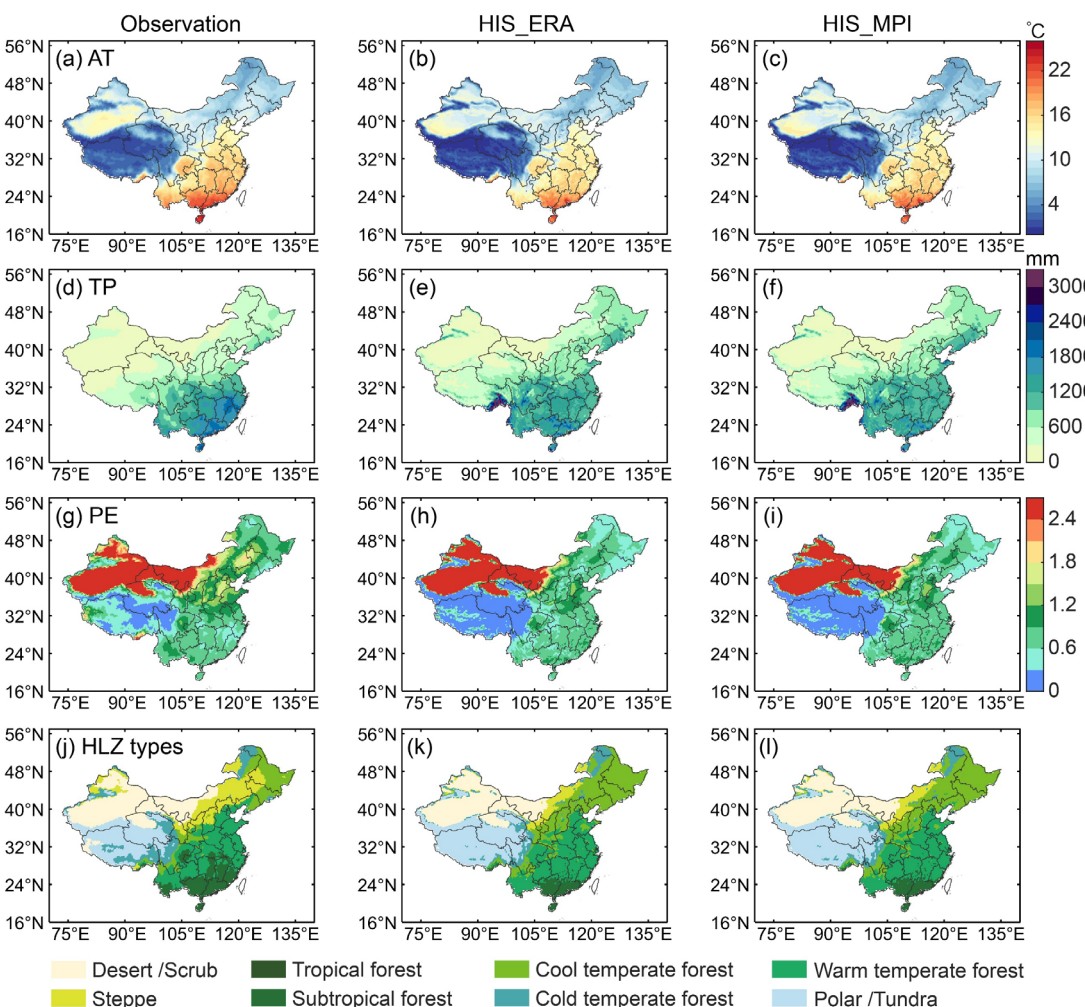

**Figure 3:** Spatial pattern of annual average biotemperature (AT), annual total precipitation (TP), potential evapotranspiration ratio (PE), and potential vegetation domain from Holdridge life zone (HLZ) model based on the observation (left), HIS_ERA simulation (center), and HIS_MPI simulation (right) during the period of 1995–2014.

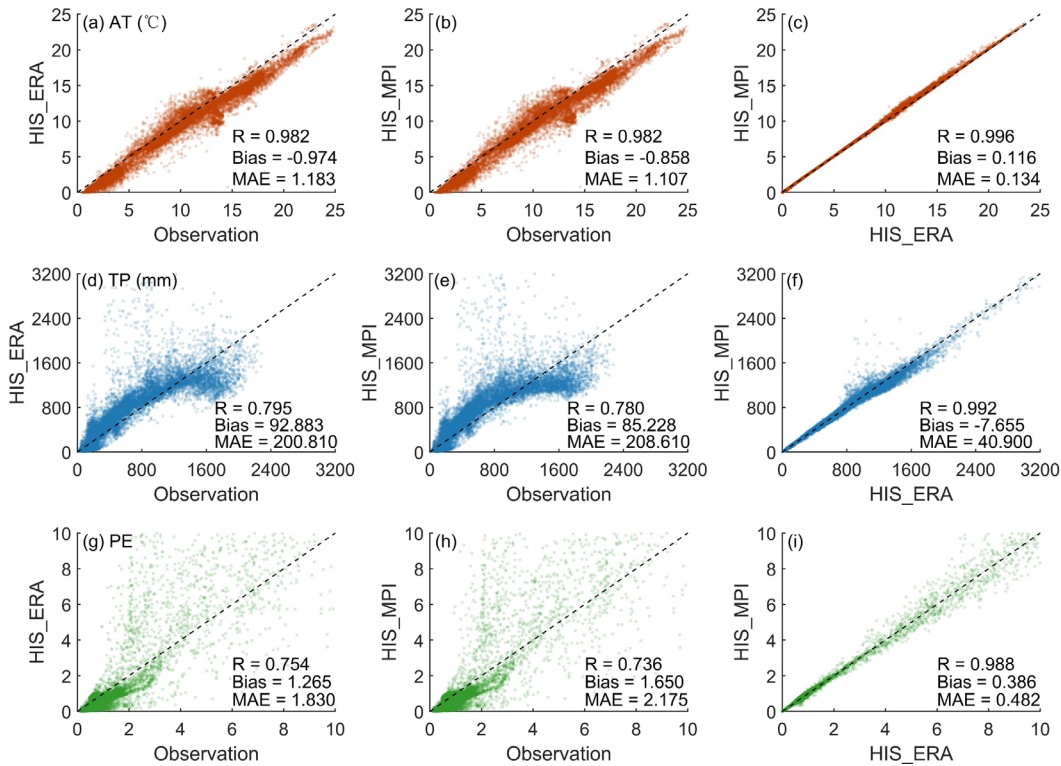

**Figure 4:** Scatterplots of the annual average biotemperature (AT), annual total precipitation (TP), and potential evapotranspiration ratio (PE) for each grid against the observation and HIS_ERA, observation and HIS_MPI, HIS_MPI and HIS_ERA. HIS_ERA and HIS_MPI indicate the WRF simulation driven by ERA5 reanalysis data and the bias-corrected MPI–ESM1–2–HR model, respectively. The observation derives from the CN05.1 dataset. Evaluation indexes included the

295 bias, mean absolute error (MAE), and spatial correlation coefficient (R). The black dotted line indicates a 1:1 line.

### 3.2 Future potential vegetable cover

For the future simulation, the three key variables (AT, TP, and PE) of the HLZ model are obtained from the FUT_MPI experiment. The projected spatial distribution of PVD is presented in Fig. 5a. The most dominant vegetation types are forest, polar/tundra, and desert/scrub, accounting for 57.1 %, 20.1 %, and 17.7 % of the total area of China, respectively. The forest

types are located in eastern China, characterized by the latitudinal direction distribution. The potential forest types from north to south are mainly cool temperate forests, warm temperate forests, and subtropical forests, in that order. It could be explained that temperature is the critical factor in defining the forest types due to the sufficient precipitation in eastern China.

Flow diagrams are useful tools for precise changes in vegetation types, displaying whether the vegetation types are shifting and in which direction. The projected changes in the area for the vegetation types are shown in Fig. 5b. The results indicate

that under future climate change, the PVD changes correspondingly. The total area of 10.4 % will be shifted in China. The northward expansion of subtropical forests replaces warm temperate forests, with an area of approximately $30.6 \times 10^4$ km$^2$,

considered the largest shifted type (Fig. S8). In addition, projected future increases in temperature and precipitation have caused some non-forestland areas to transition into forested lands (Fig. S9). For example, in western China, areas that are polar/tundra and steppe at present have transitioned into cold temperate forest and cool temperate forest in the future, respectively, and the shifted area is $18.4 \times 10^4$ km$^2$ and $1.7 \times 10^4$ km$^2$, respectively.

Overall, the PFD (1995–2014) covers approximately $500.75 \times 10^4$ km$^2$. It is projected to expand to $518.25 \times 10^4$ km$^2$, experiencing an increase of around $17.5 \times 10^4$ km$^2$ (about 3.49 %) within the 2041–2060 under the SSP2–4.5 scenario. In eastern China, the main transition is interconversions between forest types. In western China, some non-forestland types turn into forest types. These changes indicate that the forest suitable region would be changed under future climate change, and it is necessary to consider the climatic contexts in terms of future large-scale afforestation.

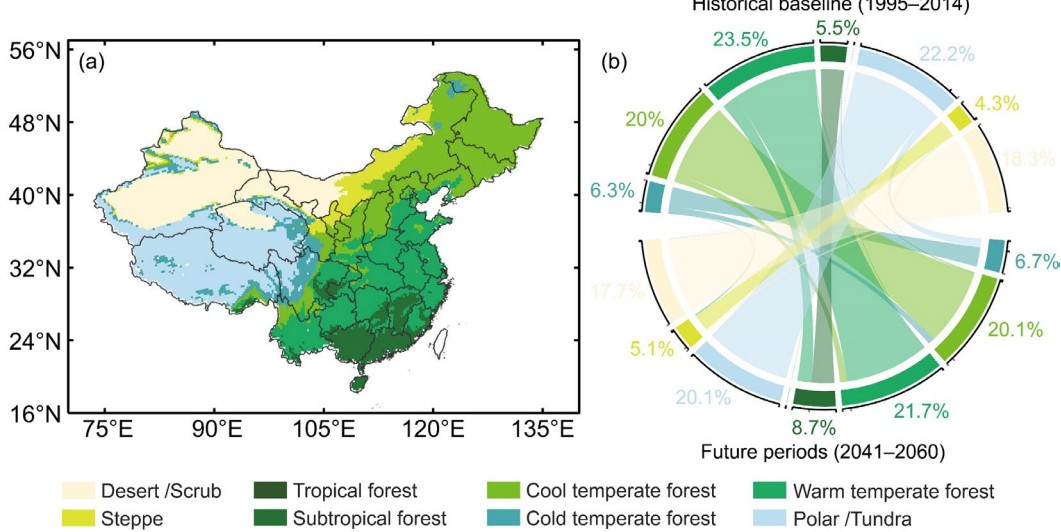

**Figure 5:** Projected spatial pattern of (a) potential vegetable types from HLZ model under the SSP2–4.5 scenario in the future period (2041–2060) from the FUT_ MPI simulation, and (b) area changes across historical baseline (1995–2014) and future period, where the calculations are based on FUT_MPI simulation versus HIS_MPI simulation.

### 3.3 Identification of future potential afforestation location in China

According to the approach of the newly afforestation allocation in section 2.2.3, we mapped the future afforestation distribution of China. First, historical open space regions for afforestation were identified. We excluded some ineligible regions, including present forestland, cropland, urban, wetland, and water bodies based on the 2020 MCD12Q1 land cover data (Fig. S1), and the remaining regions had been considered as open space regions for afforestation (Rohatyn et al., 2022). The results show that the total area of open space regions is about $612.88 \times 10^4$ km$^2$ in China, with the majority located in southern and western China (Fig. 6a).

The second step was to determine the distribution of future PFD. We used the map of potential vegetables derived from the outputs of the HLZ model (Fig. 5a) to select the forest types grids as future PFD under the SSP2–4.5 scenarios during 2041–

2060. The future PFD was considered as the forest suitable lands constrained by future climate conditions. The forest suitable

lands are mainly located in eastern China (Fig. 6b). The corresponding annual total precipitation is over 353.6 mm among the selected grids.

Then, we combined the historical open space region (Fig. 6a) with the future PFD (Fig. 6b). It enables us to obtain the future potential afforestation areas (Fig. 6c). These regions provide suitable climate conditions for forest growth and can be utilized for afforestation implementation in the context of future climate change. The total area of potential afforestation regions is

approximately $191.33 \times 10^4$ km$^2$.

There is no doubt that the potential afforestation area is extensive and unrealistic. Thus, according to the national tree planning policy, we further restricted the afforestation area. The NFMP released by the State Forestry Administration of China (2016) included the total area of planning afforestation in each province during 2020–2050 (Fig. 6d), and was considered a reference for future afforestation design. It notes that the potential afforestation area for individual provinces is usually larger than the

national planned afforestation area (Table S2). Thus, we further constrained the potential afforestation areas following the HLZ value. Specifically, we sorted the HLZ value for each province on the potential afforestation region in ascending order (Fig. 6c), and the low HLZ value were allowed priority afforestation. We calculated the total afforestation area sequentially grid by grid, until it satisfied the NFMP policy requirements. The approach of total afforestation area for each province is calculated based on Eq. (7).

$$Area = \left(0.55\, N_{Woody\ savannas} + 0.80 N_{Savannas} + N_{Grasslands\ and\ croplands}\right) \times r^2 \quad (7)$$

Where $Area$ indicates the total afforestation area. $r$ indicates the spatial resolution (here, $r$ equals 25 km). $N$ indicates the amount of afforestation grids in historical land cover. The land cover types represent the area used for afforestation. Given the tree cover for woody savannas and savannas is 30–60 % and 10–30 % according to the IGBP classification scheme (Table S1), it means that approximately 45 % and 20% of the grid area for woody savannas and savannas has already been covered forests

in the historical terms, respectively. Thus, to avoid repeated afforestation, the coefficients 0.55 and 0.80 were set.

Especially, it is worth noting that the planned afforestation area is larger than the potential afforestation area in Henan and Shandong provinces (Table S2). A small amount of cropland had been scheduled for afforestation to meet the national afforestation demand. The occupied croplands are mainly located in mountain areas, where the regions are highly suitable for forest growth. With such an afforestation scenario design, $125.33 \times 10^4$ km$^2$ croplands in China are still available for cultivation.

It is also away from the protection 'red line' of $121.67 \times 10^4$ km$^2$, released by the National Land Planning Outline (2016–2030) (State Council of China, 2017).

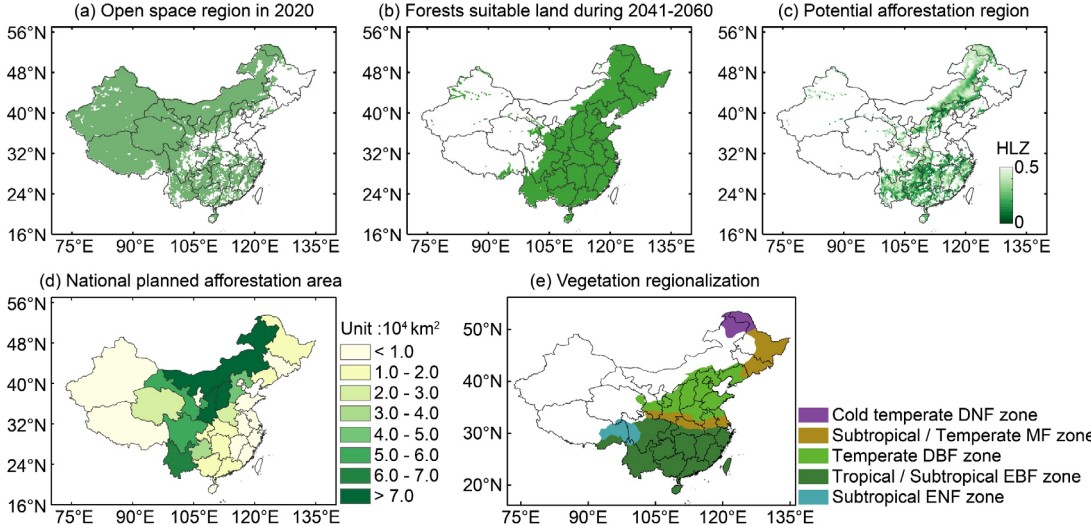

**Figure 6:** Spatial distribution of (a) historical open space region for afforestation, (b) future potential forestation domain (PFD) from HLZ model considered as the forest suitable lands, (c) potential afforestation region constrained by climate change, (d) national planned afforestation area in the individual provinces from the NFMP, (e) Chinese vegetation regionalization map.

A Chinese vegetation regionalization map (Wu et al., 1980) was used to identify the forest types within each grid (Fig. 6e). Finally, the distribution of future potential afforestation regions in China is shown in Fig. 7. The findings show that the probable locations for future potential afforestation areas in China are around and to the east of the Hu Line (a geographical division line of climate zone, and population density, economic development in China, stretching from Heihe to Tengchong). Due to afforestation, the land cover would be modified. In northern China, the main conversions types are grasslands to deciduous broadleaf forests, as well as the largest conversions in China, accounting for 40 % of the newly afforestation area. The most intensive provinces are Shanxi and Shaanxi. In southwest China, the dominant conversions are from woody savannas and savannas to evergreen broadleaf forests. These conversions account for 26 % and 16% of the newly afforestation area, respectively. These land use conversions are majorly located in southwest China, such as Yunnan province, Sichuan province, and Guizhou province. Overall, the final total afforestation area in China is approximately $73.64×10^4$ $km^2$, consistent with the NFMP ($73.78×10^4$ $km^2$). Therefore, for each province within the future afforestation region, we applied the approach mentioned above to ensure that the total afforestation area of individual provinces and extent were consistent with the national policies and future climate conditions, respectively.

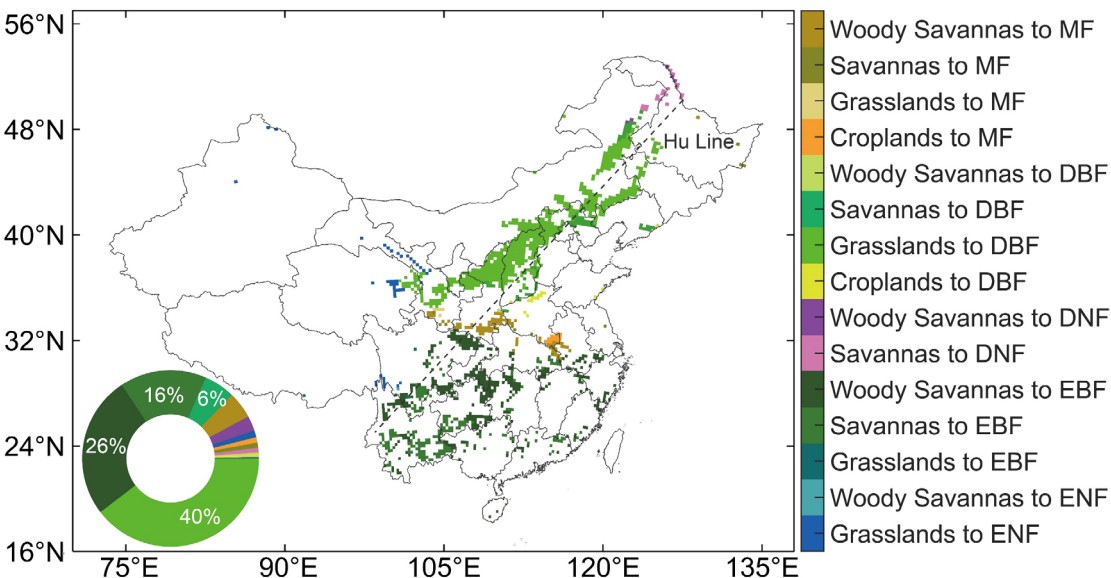

**Figure 7:** Map of future potential afforestation distribution and shift types constrained by both national afforestation plan and climate change. Forest types from IGBP include Evergreen Needleleaf Forests (ENF), Evergreen Broadleaf Forests (EBF), Deciduous Needleleaf Forests (DNF), Deciduous Broadleaf Forests (DBF), and Mixed Forests (MF). The black dotted line indicates the Hu Line.

## 4 Discussion

The most probable geographical distribution of future potential afforestation regions in China has been investigated in this study. By comparing with existing studies, the total afforestation area in this study ($73.64 \times 10^4$ km$^2$) is larger than the existing studies. For example, Zhang et al. (2022) reported an obvious increase in potential forestation lands by $33.1 \times 10^4$ km$^2$ under future climate scenarios (2070s) with the machine learning approach to predict the ecological niche of the forest. Xu (2023) reported that the area of prioritized potential forestation land was about $66.61 \times 10^4$ km$^2$ in 2020 by spatial overlay analysis of multiple factors (i.e., climate, transportation, topography, land use). However, the effects of future climate change and the national afforestation plan are ignored. Our results show that forest suitable lands will increase by $17.5 \times 10^4$ km$^2$ under the SSP2–4.5 scenario compared to the historical period. The dataset would be valuable for studying the effects of future afforestation on carbon budget, ecosystem service, water resources, and surface climate.

Our findings indicated that future afforestation in China would mostly be located around and to the east of the Hu Line, consistent with Zhang et al. (2022). The area near the Hu Line is a transition zone characterized by dry-wet, agro-pastoral, and grassland-forest. This transition zone is highly sensitive to climate change (Li et al., 2015). Due to moisture limitations, historical forest distribution is mainly located east of the Hu Line. Crossing the Hu Line is challenging for forests (Liu, 2019). However, under the future climate change, the projected results show that the temperature and precipitation in China will

increase by the middle of the 21st century under the SSP2–4.5 scenario compared to the historical period (Yang et al., 2021). A similar conclusion is also derived from our projection (Fig. S9). The response of PFD to future climate change could be slightly modified. Therefore, only a small proportion of future potential afforestation areas are in the western region of the Hu

Line, such as the Loess Plateau region. It reminds us that afforestation planning should consider vegetation responses to future climate change.

Afforestation can provide temperature benefits (e.g., cooling the land surface) according to previous studies (Peng et al., 2014; Yu et al., 2020; Breil et al., 2024). However, the biophysical response of afforestation on temperature varies spatially. At a global scale, it is common sense that afforestation causes the warming effect in high-latitude regions due to the albedo-

dominant radiation effect, while the cooling effect in low-latitude regions due to the evapotranspiration-dominant non-radiation effect (Bonan, 2008; Arora and Montenegro, 2011). Thus, afforestation-induced regional temperature changes depend on the net effects. Afforestation also can cause daytime cooling but nighttime warming (Yuan et al., 2022), and increase the surface temperature in winter, but decrease in other seasons (Ma et al., 2017). Differential responses in season and daily lead to more larger uncertainties in the net effects induced by afforestation. Therefore, a more realistic afforestation scenario is necessary

to quantify the effects of afforestation on temperature under future climate change background and develop climate change mitigation policies.

Although the resolution of our dynamical downscaled simulation (25 km) is finer than raw GCM (~100 km), it is difficult to meet the needs of afforestation planning in areas with complex topography. Convection-permitting climate modelling at the kilometre-scale has recently been developed to reproduce better mesoscale atmospheric processes (Prein et al., 2015; Lucas-

Picher et al., 2021), and obviously improve the WRF simulation, especially precipitation (Knist et al., 2020). However, improving the resolution implies higher computational costs. In contrast, statistical downscaling methods are also known to obtain high-resolution climate data with fewer computational resources (Tang et al., 2016). It assumes that the historical relationship between local climate variables and the large-scale circulation remains fixed in the future term (Wilby and Dawson, 2013). The multi-model ensemble means from the CMIP6 statistical downscaling can significantly reduce the biases compared

to individual models (Gebrechorkos et al., 2019). Thus, some statistical downscaled CMIP6 datasets (Gebrechorkos et al., 2023; Lin et al., 2023; Thrasher et al., 2022), with a resolution of 0.1°-0.25° covering the global land, can be applied to explore the future global potential afforestation area in following work. However, it is noted that the statistical downscaling data may have a limitation, as the covariance among the variables may not align with physical laws.

This study may have some limitations and uncertainties. Following the approach of existing studies (Ma et al., 2023; Qiu et

al., 2022), we also utilized the bias-correction LBCs in dynamical downscaling. However, the model uncertainty in the future climate projection is difficult to quantify because one GCM is used to nest into the WRF model. The projected result generally exhibits variations based on the choice of driving GCMs (Gao et al., 2022). This divergence can be attributed to the inherent configurations and physics parameterization of the GCM, distinct radiative forcing scenarios, and varying equilibrium climate sensitivities found in CMIP6 models (Zuo et al., 2023; Bukovsky and Mearns, 2020). For instance, the high emission scenario

could lead to higher temperature and stronger precipitation in China relative to middle emission (Yang et al., 2021). The

obvious differences are found in the northern China. It implies that there are greater opportunities for afforestation in semi-arid areas. Thus, the suitability of future forest lands depends on emission scenarios (Liu et al., 2020b; Elsen et al., 2022). Exploring the impacts of different SSPs on the distribution of potential afforestation regions would be an intriguing avenue for future research. To address the concerns about model uncertainty, exploring WRF forced by multiple bias-corrected CMIP6 models can help uncover the source of uncertainty. Utilizing ensemble means for downscaled climate simulation would contribute to a more robust projection. Additionally, the selection of different physics parameterization schemes in the WRF model can also influence the simulation performance (Gbode et al., 2019). Selecting the most suitable combination is beneficial to reduce the underlying bias. Out of all the factors limiting afforestation allocation, we used the HLZ value to constrain the afforestation distribution. Previous studies found that precipitation was a key meteorological factor that restricts forest distribution, especially in the mid-latitude regions (Hansen et al., 2005; Fang et al., 2005). If the areas with high precipitation were allowed priority afforestation, we obtained a similar future potential afforestation distribution (Fig. S10). Future studies should comprehensively consider additional factors, such as local economic development, soil physicochemical properties, and provincial tree planning policy.

**5 Conclusions**

This study evaluated the performance of the WRF model in simulating the PVD from the HLZ model in China during the historical period (1995–2014). The projected shifts in the potential vegetable types were explored under the SSP2–4.5 scenario during the future period (2041–2060) relative to the historical period. Based on these data, the most probable distribution of future potential afforestation was obtained by constraining both future climate contexts and national afforestation plans in China. We could draw the main conclusions as follows:

The output of the WRF model forced by the ERA5 analysis and bias-corrected MPI–ESM1–2–HR model could capture the spatial distribution of the PVD from the HLZ model over China through comparisons with CN05.1 dataset during the historical period. However, the WRF simulation did not precisely reproduce the observed extent of steppe types in northeast China and subtropical forests in southern China. Such misclassifications might be attributed to the bias of the precipitation simulation. Overall, in terms of the nationwide potential forestation domain, the WRF model could reproduce the spatial distribution well over China.

Under the SSP2–4.5 scenario, the PVD would obviously shift during 2041–2060 compared to the historical period. The largest shifted type was warm temperate forests to subtropical forests over southern China. The new forest suitable lands would increase by about $17.5 \times 10^4$ km$^2$ in China due to projected increased in temperature and precipitation. In addition, considering both the future climate change and national tree planning policy, we found that the probable locations for future afforestation were around and to the east of the Hu Line, with a total area of approximately $73.64 \times 10^4$ km$^2$. The main shift types were grasslands to deciduous broadleaf forests in northern China, woody savannas, and savannas to evergreen broadleaf forests in

southwest China. The findings of this study could provide a dataset for exploring the effects of future afforestation, and this method can guide designing future gridded afforestation regions for other countries.

## Data availability

The MPI–ESM1–2–HR model (Müller et al., 2018) can be downloaded from: https://esgf-ode.llnl.gov/search/cmip6/. The WRF model (Skamarock et al., 2019) can be found at: https://www2.mmm.ucar.edu/wrf/users/.National planned afforestation area data (State Forestry Administration of China, 2016) is available at: https://www.gov.cn/xinwen/2016-7/28/5095504/files/b9ac167edfd748dc8c1a256a784f40d5.pdf. The ERA5 reanalysis data (Hersbach et al., 2020) can be found at: https://cds.climate.copernicus.eu/cdsapp#!/dataset/reanalysis-era5-pressure-levels?tab=form. Chinese vegetation

regionalization map data (Wu et al., 1980) is available at: https://www.resdc.cn/data.aspx?DATAID=133. MCD12Q1 land use data (Friedl et al., 2010) can be obtained from: https://e4ftl01.cr.usgs.gov/MOTA/MCD12Q1.061/. The observed temperature and precipitation data from CN05.1 (Wu and Gao, 2013) are available at: https://ccrc.iap.ac.cn/resource/detail?id=228. The MATLAB (version 2020a) can be accessed at: https://www.mathworks.com/login?uri=%2Fdownloads%2Fweb_downloads. The future potential afforestation distribution data is available at https://zenodo.org/records/10900150.

**Author contributions**

XZ and XY designed the experiments and developed the study; SS conducted the analysis and prepared the figures. All three authors contributed to writing and revision of the text.

## Competing interests

The authors declare that they have no conflict of interest

**Acknowledgements**

This study has been supported by the National Key Research and Development Program of China (No. 2019YFA0606600 and No. 2019YFA0606904).

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
