# Peer review of "Mapping the Future Afforestation Distribution of China Constrained by National Afforestation Plan and Climate Change"

_Biogeosciences, 2023_

## Author Comment (AC1)

Response to Reviewer Comments

**General comments**

**The article "Mapping the Future Afforestation Distribution of China Constrained by National Afforestation Plan and Climate Change" explored the distribution of future potential afforestation areas based on future high-resolution climate data from the WRF model and HLZ model. It is highlighted that the afforestation scenario is constrained by both the climatological suitability for tree and national afforestation plan. The climatology suitability for tree is decided by future climate conditions and determines the potentially available afforestation domain. The national afforestation plan determines the total afforestation area. The potential value is to provide the design framework for locations of future afforestation. Overall, the article is suitable for the scope of Biogeosciences, I recommend that the authors address the concerns below in a minor revision prior to publication**.

**Response:** Thank you for your help in improving this manuscript. These comments are valuable and very helpful for revising and improving our paper. We have studied the comments and have made revisions carefully. I hope these major revisions meet with approval. The point-by-point responses to the reviewer's comments are as follows:

**Specific comments**

**Method: I'm confused about the spatial resolution of the article and please provide an explanation. Firstly, the authors emphasize the "high-resolution simulations" in this article. However, the spatial resolution is only 25 km. The other high-resolution climate dataset product (i.e., WorldClim data, https://www.worldclim.org/data/index.html.) is available at the ~1km spatial resolution. I'm confused if that description is appropriate, and please illustrate the advantages of WRF simulation in this study. L115: Why the spatial resolution of ERA5 reanalysis data is 1.0°×1.0°. In ECMWF, the highest resolution of the ERA5 product is 0.25°×0.25°, which is close to WRF simulation (25 km). In the HIS_ERA experiment, is downscaling 1.0° ERA5 data to 25 km necessary? L89: The spatial resolution of MCD12Q1 is 500m, which is different from the WRF simulation (25km). How do you match it well? Please give some detailed information.**

**Response:** Thank you for your suggestion. In this study, the spatial resolution of the WRF simulation is 25- by 25-km, which is higher than the raw CMIP6 model, ranging from 2.8125°×2.8125° (CanESM5 model) to 0.6667°×0.5° (INM-CM5-0 model). This is the meaning of high-resolution simulation in this paper. Additionally, the WorldClim data is spatially interpolated global climate data, with a spatial resolution of 1- by 1-km. Compared to the WorldClim data, the dynamical downscaling climate data (i.e., the WRF model output) has the advantage of keeping physical consistency constraints among these variables such as the hydrostatic equilibrium and the geostrophic wind balance. The physically consistent variables are an important basis for this study.

Second, on the behalf of ERA5 reanalysis data, there are actually multiple resolutions of datasets from the ECMWF. The aim of downscaled ERA5 is to evaluate the accuracy of the downscaled MPI–ESM1–2–HR model. In order to enhance the comparability of downscaled ERA5 reanalysis data and the MPI–ESM1–2–HR model, we used ERA reanalysis data with a grid size closer to that of the MPI–ESM1–2–HR model.

Third, in this study, we filled the 500-meter resolution of the MODIS data into the 25-km resolution of the WRF model grids by aggregating the MODIS pixels within a 25x25 km grid cell and calculating the area fraction of each land use type within the 25x25 km grid cell.

**L218: "Areas with high precipitation are allowed priority afforestation." In this study, precipitation is treated as a key meteorological factor that restricts forest distribution. Indeed, precipitation is critical for forest growth. However, a single climate variable is slightly simple rather than representing climatology suitability for tree. Multivariate comprehensive indicators affecting forest growth are more appropriate. In this study, the essence of the HLZ model is the distance to the three bioclimatic variables. I recommend considering the distance as a comprehensive indicator to quantify the climatology suitability for tree.**

**Response:** We agree with this comment. In the revised manuscript, the HLZ value as a comprehensive indicator has been used to quantify the climatology suitability for afforestation. Areas with a low HLZ value are allowed priority afforestation. Because a low HLZ value means a greater opportunity to be potential forestlands. Following this new method, we find that the probable locations for future potential afforestation areas in China are around and to the east of the Hu Line.

[Figure]

Figure 1: Map of future potential afforestation distribution under constraining of national afforestation planning total area and future climate changes and the afforestation-induced vegetation types conversions. Forest types from IGBP include Evergreen Needleleaf Forests (ENF), Evergreen Broadleaf Forests (EBF), Deciduous Needleleaf Forests (DNF), Deciduous Broadleaf Forests (DBF), and Mixed Forests (MF). The black dotted line indicates the Hu Line.

**L204: In the section on the approach of the newly afforestation allocation, I'm confused about the definition of forest. Please clarify it. For the national afforestation plan (NFMP), the total afforestation area is 73.78×10⁴ km². How to define the total afforestation area? I wonder whether the definition from the State Forestry Administration of China agrees with this study.**

**Response:** This study remains consistent with the definition of forest of China's State Forestry Administration. In the statistical context of China's State Forestry Administration, the term "forest area" is essentially synonymous with "woodland area." This designation is based on the criterion that the fraction of tree canopy cover exceeds 20%. The total afforestation area of 73.78×10⁴ km² implies that we will be planting trees across this area. It is anticipated that the trees would grow in health and the fraction of tree canopy cover could exceed 20%. Thereby, afforestation implicates that non-woodland would be replaced by woodland. Given the need to ensure food security, urban expansion, and ecological protection in the future, future afforestation cannot occupy cropland, urban, and wetlands (including water bodies). Therefore, the implementation of future afforestation in this study occurs mainly on the present grassland, savanna, and woody savanna.

**L113: The authors use the SSP2–4.5 scenario (the middle-of-the-road development) to represent the climate future projections. However, this study only used one model projections rather than multiple model ensemble mean. Following the methodology of CMIP6 climate projection, scenario-based climate projection may have large uncertainties. It is suggested the revision to address this issue. It is also worthy to discuss effects of single model projection uncertainties on the research result of this study.**

**Response:** Following this comment, we revised the discussion. In the revised manuscript, the model scenarios and uncertainties are discussed, as follows:

This study may have some limitations and uncertainties. Following the approach of existing studies (Ma et al., 2023; Qiu et al., 2022), we also utilized the bias-correction LBC in dynamical downscaling. However, the model uncertainty in the future climate projection is difficult to quantify because one GCM is used to nest into the WRF model. The projected result generally exhibits variations based on the choice of driving GCMs (Gao et al., 2022). This divergence can be attributed to the inherent configurations and physics parameterization of the GCMs, distinct radiative forcing scenarios, and varying equilibrium climate sensitivities found in CMIP6 models (Zuo et al., 2023; Bukovsky and Mearns, 2020). For instance, the high emissions scenario could lead to higher temperature and stronger precipitation in China (Yang et al., 2021). Consequently, the suitability of land for future forests may change accordingly. Exploring the impacts of different SSPs on the distribution of potential afforestation regions would be an intriguing avenue for future research.

To address the concerns about model uncertainty, exploring WRF forced by multiple bias-corrected CMIP6 models can help uncover the source of uncertainty. Utilizing ensemble means for downscaled climate simulation would contribute to a more robust projection. Additionally, the selection of different physics parameterization schemes in the WRF model can also influence the simulation performance (Gbode et al., 2019). Selecting the most suitable combination is beneficial to reduce the underlying bias.

Although the resolution of our dynamical downscaled simulation (25 km) is finer than raw GCMs (~100 km), it is difficult to meet the needs of afforestation planning in areas with complex topography. Convection-permitting climate modelling at the kilometre-scale has recently been developed to reproduce better mesoscale atmospheric processes (Prein et al., 2015; Lucas-Picher et

al., 2021), and obviously improve the WRF simulation, especially precipitation (Knist et al., 2020).

However, increasing the resolution of the simulation implies higher computational costs. In contrast, statistical downscaling methods are also known to obtain high-resolution climate data with fewer computational resources (Tang et al., 2016). It assumes that the historical relationship between local climate variables and the large-scale circulation remains fixed in the future term (Wilby and Dawson, 2013). The multi-model ensemble means from the CMIP6 statistical downscaling can significantly reduce the biases compared to individual models (Gebrechorkos et al., 2019). Thus, some statistical downscaled CMIP6 datasets (Gebrechorkos et al., 2023; Lin et al., 2023; Thrasher et al., 2022), with a resolution of 0.1°-0.25° covering the global land, can be applied to explore the future global potential afforestation area in following work. However, it is noted that the statistically downscaling data may have a limitation, as the covariance among the variables may not align with physical laws.

References

Tang, J., Niu, X., Wang, S., Gao, H., Wang, X., & Wu, J. (2016). Statistical downscaling and dynamical downscaling of regional climate in China: Present climate evaluations and future climate projections. Journal of Geophysical Research: Atmospheres, 121(5), 2110-2129.

Ahmed, K. F., Wang, G., Silander, J., Wilson, A. M., Allen, J. M., Horton, R., & Anyah, R. (2013). Statistical downscaling and bias correction of climate model outputs for climate change impact assessment in the US northeast. Global and Planetary Change, 100, 320-332.

Gebrechorkos, S., Hülsmann, S., & Bernhofer, C. (2019). Regional climate projections for impact assessment studies in East Africa. Environmental Research Letters, 14(4), 044031.

Gebrechorkos, S., Leyland, J., Slater, L., Wortmann, M., Ashworth, P. J., Bennett, G. L., et al. (2023). A high-resolution daily global dataset of statistically downscaled CMIP6 models for climate impact analyses. Scientific Data, 10(1), 611.

Lin, H., Tang, J., Wang, S., Wang, S., & Dong, G. (2023). Deep learning downscaled high-resolution daily near surface meteorological datasets over East Asia. Scientific Data, 10(1), 890.

Thrasher, B., Wang, W., Michaelis, A., Melton, F., Lee, T., & Nemani, R. (2022). NASA global daily downscaled projections, CMIP6. Scientific Data, 9(1), 262.

Wilby, R. L., & Dawson, C. W. (2013). The statistical downscaling model: insights from one decade of application. International Journal of Climatology, 33(7), 1707-1719.

Gbode, I. E., Dudhia, J., Ogunjobi, K. O., & Ajayi, V. O. (2019). Sensitivity of different physics schemes in the WRF model during a West African monsoon regime. Theoretical and Applied Climatology, 136, 733-751.

Prein, A. F., Langhans, W., Fosser, G., Ferrone, A., Ban, N., Goergen, K., et al. (2015). A review on regional convection-permitting climate modeling: Demonstrations, prospects, and challenges. Reviews of Geophysics, 53(2), 323-361.

Lucas-Picher, P., Argüeso, D., Brisson, E., Tramblay, Y., Berg, P., Lemonsu, A., et al. (2021). Convection-permitting modeling with regional climate models: Latest developments and next steps. Wiley Interdisciplinary Reviews: Climate Change, 12(6), e731.

Knist, S., Goergen, K., & Simmer, C. (2020). Evaluation and projected changes of precipitation statistics in convection-permitting WRF climate simulations over Central Europe. Climate Dynamics, 55(1-2), 325-341.

Ma, M., Tang, J., Ou, T., & Zhou, P. (2023). High-resolution climate projection over the Tibetan Plateau using WRF forced by bias-corrected CESM. Atmospheric Research, 286, 106670.

Qiu, Y., Feng, J., Yan, Z., Wang, J., & Li, Z. (2022). High-resolution dynamical downscaling for regional climate projection in Central Asia based on bias-corrected multiple GCMs. Climate Dynamics, 58(3-4), 777-791.

Gao, S., Zhu, S., & Yu, H. (2022). Dynamical downscaling of temperature extremes over China using the WRF model driven by different lateral boundary conditions. Atmospheric Research, 278, 106348.

Zuo, Z., Fung, J. C., Li, Z., Huang, Y., Wong, M. F., Lau, A. K., & Lu, X. (2023). Projection of future heatwaves in the Pearl River Delta through CMIP6-WRF dynamical downscaling. Journal of Applied Meteorology and Climatology, 62(9), 1297-1314.

Bukovsky, M. S., & Mearns, L. O. (2020). Regional climate change projections from NA-CORDEX and their relation to climate sensitivity. Climatic Change, 162(2), 645-665.

Yang, X., Zhou, B., Xu, Y., & Han, Z. (2021). CMIP6 evaluation and projection of temperature and precipitation over China. Advances in Atmospheric Sciences, 38, 817-830.

**L353: "Our findings indicated that future afforestation in China would mostly occur around and to the east of the Hu Line, consistent with Zhang et al. (2022)." The authors try to compare other similar studies on future potential afforestation distribution. More result differences should be discussed. I suggest to highlight the innovation and implications of the article by comparing with existing studies.**

**Response:** In the revision, we discussed the innovation and implications of the article by comparing with existing studies, as follows:

The most probable geographical distribution of future potential afforestation regions in China has been investigated in this study. Compared with previous studies, the total afforestation area in this study is greater than theirs. For example, Zhang et al. (2022) reported future climate changes may lead to an increase in suitable forestation lands by $33.1 \times 10^4$ km$^2$ (2070s) through predicting the ecological niche of the forest using the machine learning approach. Xu et al. (2023) found that the area of prioritized potential forestation land was about $66.61 \times 10^4$ km$^2$ in 2020 through spatial overlay analysis by considering multiple factors including climate, transportation, topography, land use and so on. This study is oriented towards national afforestation plans to identify future potential afforestation regions referring to climate change scenarios and land use patterns. The overall result is a more realistic and plausible afforestation scenario. The dataset would be valuable for studying the effects of future afforestation on carbon budget, ecosystem service, water resources, and surface climate.

References

Zhang, L., Sun, P., Huettmann, F., & Liu, S. (2022). Where should China practice forestry in a warming world?. Global Change Biology, 28(7), 2461-2475.

Xu, J. (2023). Estimation of the spatial distribution of potential forestation land and its climatic potential productivity in China. Acta Geographica Sinica, 78(3), 677–693.

**L180-186: Why the Holdridge life zone (HLZ) model is suitable for simulating the potential vegetation types in China. The author simply describes the extensive application of the HLZ model. I suggest validating the accuracy of the HLZ model. It is necessary to compare potential vegetation types with true vegetation types. Please add it to the Supplement Material.**

**Response:** Following this suggestion, we included a comparison between potential vegetation from the HLZ model and the actual vegetation. The actual vegetation in the year 2005 refers to China's Land-Use/cover Datasets (CLUDs), which has a spatial resolution of 1- by 1 km and covers the entire China (Liu et al., 2014). The overall accuracy of CLUDs is above 90% (Liu et al., 2010). We find that the HLZ model can reproduce the potential forest distribution and grassland-forest geographical boundary well.

Reference

Liu, J., Kuang, W., Zhang, Z., Xu, X., Qin, Y., Ning, J., et al. (2014). Spatiotemporal characteristics, patterns, and causes of land-use changes in China since the late 1980s. Journal of Geographical Sciences, 24, 195-210.

Liu, J., Zhang, Z., Xu, X., Kuang, W., Zhou, W., Zhang, S., et al. (2010). Spatial patterns and driving forces of land use change in China during the early 21st century. Journal of Geographical Sciences, 20, 483-494.

[Figure]

Figure 2. Comparison of actual (a) and potential (b) nature vegetation types. The actual vegetation types refer to China's Land-Use/cover Datasets (CLUDs) for the year 2005. The potential vegetation types derived from the HLZ model are based on the average of 1995-2014.

**L125: The authors have done substantial work on numerical experiments. For example, the authors correct the lateral boundary conditions rather than the raw GCM before dynamic downscaling. It is a very good solution to reduce the underlying bias. I suggest adding the**

**comparison of raw GCM, bias-corrected GCM, and observation**.

**Response:** To select the excellent performance of GCM, our previous (Song et al., 2023) studies comprehensively evaluated the performance of the GCM involved in CMIP6. It was reported that the MPI–ESM1–2–HR model from the Max Planck Institute outperforms all other GCMs in East Asia. We have added the comparison of the raw MPI–ESM1–2–HR model, bias-corrected MPI–ESM1–2–HR model, and the observation. The bias-corrected MPI–ESM1–2–HR model can reduce the underlying biases. The results are as follows:

[Figure]

Figure 3: Comparison of ERA5 reanalysis data with raw (No_BC_MPI) and bias-corrected historical MPI–ESM1–2–HR model (BC_MPI) at the pressure of 850 hPa for the period 1995–2014.

The odd rows represent the spatial distribution of climatology, and the even rows represent the differences.

[Figure]

Figure 4: Comparison of ERA5 reanalysis data with raw (No_BC_MPI) and bias-corrected historical MPI–ESM1–2–HR model (BC_MPI) at the different pressure levels and months.

Reference

Song, S., Zhang, X., Gao, Z., & Yan, X. (2023). Evaluation of atmospheric circulations for dynamic downscaling in CMIP6 models over East Asia. Climate Dynamics, 60(7-8), 2437-2458.

**L351: This article emphasizes "The dataset would be valuable for studying the effects of future afforestation on carbon budget, ecosystem service, water resources, surface climate". Would the data set be available to the public, especially in Figure 7**?

**Response:** Yes. This dataset is available from the corresponding author upon request.

**L234: "The WRF simulation generally overestimates TP in most regions with a national-average bias of 92.883 mm". According to Figure 3d-3f, the obvious overestimate is over the southeast Tibetan Plateau. It is suggested to explain the potential reasons of these bias in the revision**.

**Response:** The southeastern Tibetan Plateau (TP) is characterized by complex terrain. Regional climate models (RCMs) generally overestimate precipitation over the TP (Wang et al., 2021; Liu, et al., 2023). The wet bias could be attributed to inappropriate parameterization schemes (Ou et al., 2020; Zhao et al., 2023), coarse horizontal resolution (Lin et al., 2018; Rahimi et al., 2019), and inappropriate land-surface processes associated with soil moisture and frozen–thawing (Fu et al., 2020; Yang et al., 2018). For example, a high-resolution simulation can reproduce more realistic terrain characteristics and reduce the wet bias because finer resolutions decrease the water vapour transport towards the TP due to improving resolving orographic drag (Lin et al., 2018). An improved cloud macrophysics scheme can increase low cloud cover and reduce latent heat flux and land surface temperature, which leads to a more stable atmosphere and less precipitation (Zhao et al., 2023).

References

Lin, C., Chen, D., Yang, K., & Ou, T. (2018). Impact of model resolution on simulating the water vapor transport through the central Himalayas: implication for models' wet bias over the Tibetan Plateau. Climate Dynamics, 51, 3195-3207.

Zhao, D., Lin, Y., Dong, W., Qin, Y., Chu, W., Yang, K., et al. (2023). Alleviated WRF summer

wet bias over the Tibetan Plateau using a new cloud macrophysics scheme. Journal of Advances in Modeling Earth Systems, 15(10), e2023MS003616.

Ou, T., Chen, D., Chen, X., Lin, C., Yang, K., Lai, H. W., & Zhang, F. (2020). Simulation of summer precipitation diurnal cycles over the Tibetan Plateau at the gray-zone grid spacing for cumulus parameterization. Climate Dynamics, 54, 3525-3539.

Rahimi, S. R., Wu, C., Liu, X., & Brown, H. (2019). Exploring a variable-resolution approach for simulating regional climate over the Tibetan Plateau using VR-CESM. Journal of Geophysical Research: Atmospheres, 124(8), 4490-4513.

Fu, Y., Ma, Y., Zhong, L., Yang, Y., Guo, X., Wang, C., et al. (2020). Land-surface processes and summer-cloud-precipitation characteristics in the Tibetan Plateau and their effects on downstream weather: a review and perspective. National Science Review, 7(3), 500-515.

Yang, K., Wang, C., & Li, S. (2018). Improved simulation of frozen-thawing process in land surface model (CLM4. 5). Journal of Geophysical Research: Atmospheres, 123(23), 13-238.

Wang, X., Tolksdorf, V., Otto, M., & Scherer, D. (2021). WRF-based dynamical downscaling of ERA5 reanalysis data for High Mountain Asia: Towards a new version of the High Asia Refined analysis. International Journal of Climatology, 41(1), 743-762.

Liu, H., Zhao, X., Duan, K., Shang, W., Li, M., & Shi, P. (2023). Optimizing simulation of summer precipitation by weather research and forecasting model over the mountainous southern Tibetan Plateau. Atmospheric Research, 281, 106484.

**Table 1: Why this parameterization scheme of the WRF model is appropriate in this study. Please give a specific reason or reference**.

**Response:** The reference was included in the revision.

Reference

Hu, Y., Zhang, X. Z., Mao, R., Gong, D. Y., Liu, H. B., & Yang, J. (2015). Modeled responses of summer climate to realistic land use/cover changes from the 1980s to the 2000s over eastern China. Journal of Geophysical Research: Atmospheres, 120(1), 167-179.

**L300: What is the meaning of "The corresponding annual total precipitation is over 353.6 mm among the selected grids"? How to obtain the value of 353.6 mm. Please clarify it**.

**Response:** In the HLZ model, the minimum precipitation for forests is prescribed as 353.6 mm.

In the revision, it is revised as:

Their annual precipitation is all above 353.6 mm, which is prescribed as a precipitation limitation for forests in the HLZ model.

**L311: "It is generally common sense that afforestation is highly constrained by precipitation." Please add specific explanations or references**.

**Response:** The references were included in the revision.

References

Harvey, J. E., Smiljanić, M., Scharnweber, T., Buras, A., Cedro, A., Cruz-García, R., et al. (2020). Tree growth is influenced by a warming winter climate and summer moisture availability in northern temperate forests. Global Change Biology, 26(4), 2505-2518.

Fang, J., Piao, S., Zhou, L., He, J., Wei, F., Myneni, R. B., et al. (2005). Precipitation patterns alter growth of temperate vegetation. Geophysical Research Letters, 32(21).

**L275: To what does "total area" refer to? Is it the whole nation? Please clarify**.

**Response:** The "total area" refers to the entire China land area.

**Figure 5b: The flow diagrams are not clear, and please give specific values**.

**Response:** The flow diagram was improved and specific values are included.

[Figure]

Figure 5: Projected spatial pattern of (a) potential vegetable types from HLZ model under the SSP2–4.5 scenario in the future periods (2041–2060) from the FUT_MPI simulation, and (b) area changes across historical baseline (1995–2014) and future periods, where the calculations are based on FUT_MPI simulation versus HIS_MPI simulation.

**Eq. (2): " , , and ". Please correct it.**

**Response:** The Equation (2) was corrected in the revision.

$$F_{cor} = D_{GCM\_F} \times \frac{SD_{ERA}}{SD_{GCM}} + M_{ERA} + \left( M_{GCM\_F} - M_{GCM\_H} \right) \quad (2)$$

**Figure 3 and Figure 4: For Figure 3 and Figure 4 captions, suggest not to use the abbreviations "HLZ", "AT", "TP", and "PE".**

**Response:** The full names were included in the captions.

**L208: "national afforestation plan" is redundant. Please use the "NFMP".**

**Response:** It is revised.

**L98: "The total national afforestation area is about 73.78×10⁴ km² from 2020 to 2050". Please give specific forest cover.**

**Response:** It was revised as follows:

The national afforestation plan shows a total afforestation area of about 73.78×10⁴ km²

(equivalent to an increase China's forest cover by 7.7%) from 2020 to 2050.

**Figure 2: No citation for Figure 2 in the text**.

**Response:** The citation of Figure 2 was included in the revision.

**Figure 6: Please do not use the abbreviations in the figure captions**.

**Response:** All full name is presented in the captions in revision.

**L338: "woody savannas" replaces "Woody savannas"**.

**Response:** It is revised.

---

## Author Comment (AC2)

**Response to Reviewer Comments**

**General comments:**

**This manuscript attempted to map the future afforestation distribution in China. This future afforestation distribution plays an important role in land-atmosphere interactions and carbon cycle research, but it hardly been obtained so far. The authors provided a technological roadmap to deal with it. Compared to previous idealistic and hypothetical afforestation scenarios, this study designed a plausible afforestation scenario due to considering the national afforestation plan. The study also did a relatively good job at dynamical downscaling of GCM outputs in terms of future climate projection. Overall, the study adopted a novel perspective and robust technique for identifying future potential afforestation domains.**

**I find that this paper is very intriguing and important and lots of additional work behind this study is worth further exploring. The manuscript could be accepted as I believe. On the other hand, I also have several minor comments. I hope that these comments can improve the manuscript. My comments are given below.**

**Response:** We thank you for your interest in our study. These comments are valuable and very helpful for revising and improving our paper. We have studied the comments carefully and have made revisions to the revised version. I hope these major revisions meet with approval. The point-by-point responses to the reviewer's comments are as follows:

**Specific comments:**

**Why is the SSP2–4.5 scenario selected? There are several shared socioeconomic pathways (SSPs) for future climate projections in the CMIP6. The study results may be dependent on the selection of SSPs. Why is the SSP2–4.5 scenario suitable for your studies?**

**Response:** CMIP6 used a new scenario projection framework combined with the SSPs (i.e., SSP1-2.6, SSP2-4.5, SSP5-8.5). It is indeed that projected precipitation and temperature vary across the SSPs. Thus, the future forests suitable lands may be divergent. It is reported that the middle-of-the-road development (SSP2–4.5 scenario) represented the most likely development path to occur in China. Therefore, this study used the SSP2–4.5 scenario. In future, we can further compare the effects of different SSPs on the distribution of potential afforestation regions.

**By comparing potential vegetation domain simulation with observation, some disagreement could be found. For example, in southern China, the observed subtropical forest expands northward up to 32°N. However, the simulation results reduce the extent. Given the bias in the WRF model simulation, why does this simulation still make sense?**

**Response:** In this study, the main role of the WRF simulation is to identify the extent of forest suitable land. In order to reduce the effect of the WRF simulation, this study has corrected the bias of the lateral boundary conditions. Compared with the actual forest pattern, the WRF simulation could reproduce the distribution of potential forest regions in China well. Compared to the national afforestation plan, the bias in the extent of forest suitable land due to WRF simulation has a small impact on the results of this study.

**This study only used an MPI–ESM1–2–HR model as the lateral boundary of WRF model. It may fail to obtain robust future climate projections. The NEX-GDDP-CMIP6 (NASA Earth eXchange Global Daily Downscaled Projections CMIP6 Data) datasets contain multiple GCMs and SSPs with a spatial resolution of 0.25° × 0.25°, which is approximate same with this study of 25- by 25-km. Why not use this dataset? The relevant reference is "Thrasher, B., Wang, W., Michaelis, A., Melton, F., Lee, T., & Nemani, R. (2022). NASA global daily downscaled projections, CMIP6. Scientific Data, 9(1), 262."**

**Response:** Thank you for bringing this recent study to our attention. The NEX-GDDP-CMIP6 datasets are developed by the statistical downscaling algorithm. Compared to the NEX-GDDP-CMIP6 datasets, the dynamic downscaling climate data (i.e., WRF model output) has the advantage of keeping physical consistency constraints between variables such as the hydrostatic equilibrium and geostrophic wind balance. Thus, dynamical downscaling climate data is used in this study. The discussion on the statistical downscaling was included in the revision, as follows:

Although the resolution of our dynamical downscaled simulation (25 km) is finer than raw GCMs (~100 km), it is difficult to meet the needs of afforestation planning in areas with complex topography. Convection-permitting climate modelling at the kilometre-scale has recently been developed to reproduce better mesoscale atmospheric processes (Prein et al., 2015; Lucas-Picher et al., 2021), and obviously improve the WRF simulation, especially precipitation (Knist et al., 2020).

However, increasing the resolution of the simulation implies higher computational costs. In contrast, statistical downscaling methods are also known to obtain high-resolution climate data with fewer computational resources (Tang et al., 2016). It assumes that the historical relationship between local climate variables and the large-scale circulation remains fixed in the future term (Wilby and Dawson, 2013). The multi-model ensemble means from the CMIP6 statistical downscaling can significantly reduce the biases compared to individual models (Gebrechorkos et al., 2019). Thus, some statistical downscaled CMIP6 datasets (Gebrechorkos et al., 2023; Lin et al., 2023; Thrasher et al., 2022), with a resolution of 0.1°-0.25° covering the global land, can be applied to explore the future global potential afforestation area in following work. However, it is noted that the statistically downscaling data may have a limitation, as the covariance among the variables may not align with physical laws.

References

Prein, A. F., Langhans, W., Fosser, G., Ferrone, A., Ban, N., Goergen, K., et al. (2015). A review on regional convection-permitting climate modeling: Demonstrations, prospects, and challenges. Reviews of Geophysics, 53(2), 323-361.

Lucas-Picher, P., Argüeso, D., Brisson, E., Tramblay, Y., Berg, P., Lemonsu, A., et al. (2021). Convection-permitting modeling with regional climate models: Latest developments and next steps. Wiley Interdisciplinary Reviews: Climate Change, 12(6), e731.

Knist, S., Goergen, K., & Simmer, C. (2020). Evaluation and projected changes of precipitation statistics in convection-permitting WRF climate simulations over Central Europe. Climate Dynamics, 55(1-2), 325-341.

Tang, J., Niu, X., Wang, S., Gao, H., Wang, X., & Wu, J. (2016). Statistical downscaling and dynamical downscaling of regional climate in China: Present climate evaluations and future climate projections. Journal of Geophysical Research: Atmospheres, 121(5), 2110-2129.

Wilby, R. L., & Dawson, C. W. (2013). The statistical downscaling model: insights from one decade of application. International Journal of Climatology, 33(7), 1707-1719.

Gebrechorkos, S., Hülsmann, S., & Bernhofer, C. (2019). Regional climate projections for impact assessment studies in East Africa. Environmental Research Letters, 14(4), 044031.

Gebrechorkos, S., Leyland, J., Slater, L., Wortmann, M., Ashworth, P. J., Bennett, G. L., et al. (2023). A high-resolution daily global dataset of statistically downscaled CMIP6 models for climate

impact analyses. Scientific Data, 10(1), 611.

Lin, H., Tang, J., Wang, S., Wang, S., & Dong, G. (2023). Deep learning downscaled high-resolution daily near surface meteorological datasets over East Asia. Scientific Data, 10(1), 890.

Thrasher, B., Wang, W., Michaelis, A., Melton, F., Lee, T., & Nemani, R. (2022). NASA global daily downscaled projections, CMIP6. Scientific Data, 9(1), 262.

**From Figure 6c to Figure 7, you further constrained the afforestation area through the total precipitation. Precipitation is important but not the only determinant of afforestation allocation. More other factors may be needed to be considered.**

Response: We agree with this comment. In the revised manuscript, the HLZ value as a comprehensive indicator has been used to quantify the climatology suitability for afforestation. Areas with a low HLZ value are allowed priority afforestation. Because a low HLZ value means a greater opportunity to be potential forestlands. Following this new method, we find that the probable locations for future potential afforestation areas in China are around and to the east of the Hu Line.

[Figure]

Figure 1: Map of future potential afforestation distribution under constraining of national afforestation planning total area and future climate changes and the afforestation-induced vegetation types conversions. Forest types from IGBP include Evergreen Needleleaf Forests (ENF), Evergreen Broadleaf Forests (EBF), Deciduous Needleleaf Forests (DNF), Deciduous Broadleaf Forests (DBF), and Mixed Forests (MF). The black dotted line indicates the Hu Line.

**It's not clear that "This bias-corrected approach was applied to the variables such as air temperature, specific humidity, zonal wind, meridional wind, geopotential height, etc." in Line 121. In addition to these five meteorological variables, were there other variables bias-corrected? More detail please.**

**Response:** In fact, there are a total of 16 bias-corrected variables. These include five atmospheric fields, i.e. air temperature, specific humidity, zonal wind, meridional wind, geopotential height, and eleven surface fields, i.e. surface temperature, sea-surface temperature, surface pressure, sea ice cover, sea-level pressure, soil temperature, soil moisture, near-surface temperature, relative humidity, zonal and meridional wind.

**The authors need to add more descriptions of the future potential afforestation distribution and shift types (Figure 7). It seems that this part of the manuscript is too short.**

**Response:** We have expanded this description as follows:

The findings show that the probable locations for future potential afforestation areas in China would be around and to the east of the Hu Line. Due to afforestation, the land cover would be modified. In northern China, the dominant conversion is from grasslands to deciduous broadleaf forests. Such conversion is also the most dominant land use change due to afforestation. It accounts for 40 % of the newly afforestation area. In detail, among the provinces in northern China, the largest conversion from grassland to deciduous broadleaf forest may occur in Shanxi and Shaanxi. In southwest China, the dominant conversions are from woody savannas and savannas to evergreen broadleaf forests. These conversions account for 26 % and 16% of the newly afforestation area, respectively. These land use conversions are majorly located in southwest China, such as Yunnan province, Sichuan province, and Guizhou province.

**In line 209, here, it is stated that the cropland does not encroach on afforestation. However, Figure 7 shows that the shift types include croplands to MF and croplands to DBF. Is that a contradiction here? This should be commented.**

**Response:** We have revised the manuscript. In the revision, the criteria for identifying potential afforestation areas were changed to minimize encroachment on cropland. If the historical grassland,

savannas and woody savannas do not meet the demand of the national afforestation plan, we just consider encroachment on the cropland. The result shows that a small amount of cropland has been scheduled for afforestation to meet the national afforestation requirement. Overall, 1.88 billion mu croplands in China are still available for cultivation. It is also away from the protection 'red line' of 1.865 billion mu, released by the National Land Planning Outline (2016–2030) (State Council of China, 2017).

**In line 292, "We exclude some ineligible regions, including present forestland, cropland, urban, wetland, and water bodies based on the 2020 MCD12Q1 land cover data". This sentence is repeated. The definition of "historical open space regions" has been clarified in section 2.2.3.**

**Response:** In the revision we deleted it.

**Line 114. The presentation on the ERA5 dataset is too short. Which meteorological variables are used in the study? What is the time scale and spatial extent?**

**Response:** In the revision, we introduced more detail about EAR5 as follows:

The ERA5 reanalysis data is the fifth generation global reanalysis product developed by the European Centre for Medium-range Weather Forecast (ECMWF) (Hersbach et al., 2020). The state-of-the-art reanalysis data assimilates multi-source data including ground-based meteorological measurements data, satellite-observed data, and atmospheric sounding data based on 4D-var ensemble data assimilation system (Hersbach et al., 2020). The 6–hourly ERA5 reanalysis data with a spatial resolution of $1.0°×1.0°$ from 1994 to 2014 was also used as the lateral boundary conditions. The related meteorological variables for the MPI–ESM1–2–HR model and ERA5 reanalysis data included atmospheric fields (air temperature, specific humidity, zonal wind, meridional wind, geopotential height) and surface fields (i.e., sea-surface temperature, sea ice cover, soil temperature and soil moisture, etc.).

Reference

Hersbach, H., Bell, B., Berrisford, P., Hirahara, S., Horányi, A., Muñoz-Sabater, J., et al. (2020). The ERA5 global reanalysis. Quarterly Journal of the Royal Meteorological Society, 146(730), 1999-2049.

**Technical corrections:**

**Line 100 – "Climate Modelling" replace "Climate modelling".**

**Response:** It is revised.

**Table 1 – Give specific model top pressure.**

**Response:** We have added it. The model top pressure is 50hPa in this study.

**Line 125 – Check the Equation (2).**

**Response:** We apologize for this mistake. We have corrected the Equation (2):

$$F_{cor} = D_{GCM\_F} \times \frac{SD_{ERA}}{SD_{GCM}} + M_{ERA} + \left( M_{GCM\_F} - M_{GCM\_H} \right) \quad (2)$$

**Line 393– "Woody savannas" -> "woody savannas"**

**Response:** Done.

**Line 76 – "The fourth section will be the discussion."**

**Response:** We have refined it as follows:

The discussion and conclusions are summarized in sections four and five.

**Table 1 – This should be "Initial and lateral boundary conditions"**

**Response:** It is revised.

**Line 14 – 7. In the abstract section, the abbreviation (WRF) should be the full name. Make sure the reader understandings.**

**Response:** All full name is presented in the captions in revision.

**Table 1 – "ERA5 analysis" -> "ERA5 reanalysis"**

**Response:** Done.

**Line 102– "CMIP6". Add full name.**

**Response:** The full name is the Coupled Model Intercomparison Project 6 (CMIP6).

**Line 17– "SSP". Add full name.**

**Response:** The full name is the shared socioeconomic pathways (SSP).

**Line 20– "occur" -> "be located"**

**Response:** It is revised.

**Line 54– "employ" -> "employed"**

**Response:** Done.

**Line 74– "the total area afforestation" -> "the total afforestation area"**

**Response:** Done.

**Line 83– "from 1995–2014" -> "from 1995 to 2014"**

**Response:** It is revised.

**Line 89– "features" -> "featured"**

**Response:** Done.

**Line 378– "historical periods" -> "historical period"**

**Response:** Done.

---

## Author Comment (AC3)

**Response to Reviewer Comments**

**In this manuscript, Song, Zhang, and Yan mapped the future afforestation distribution of China under political guidance and climate change. It is a good example to serve the society using numerical techniques. Overall, this manuscript is clear-written, easy to understand, and seems to be methodologically sound. I like the most of plots in this manuscript. However, I still have several comments that should be addressed below.**

**Response:** We thank the reviewer for all the constructive comments provided. These comments are valuable and very helpful for revising and improving our paper. We have studied the comments carefully and have made revisions to the revised version. I hope these major revisions meet with approval. The point-by-point responses to the reviewer's comments are as follows:

**Specific comments:**

**Line 20: Please explain the Hu Line. Readers outside are not familiar with this geographical division.**

**Response:** We have added the explanation in the abstract as follows:

The newly afforestation grid cells would be located around and to the east of the Hu Line (a geographical division line stretching from Heihe to Tengchong).

We have added the details in the results as follows:

Hu Line, a geographical division line of climate zone, and population density, economic development in China, stretches from Heihe to Tengchong.

**Line 24: Replace "surface climate" with "surface hydroclimate regime".**

**Response:** It is revised.

**Line 29: Afforestation not only influences the land surface energy and mass budgets, but also affects the water cycle. Water cycle should be mentioned, since in the main text PRCP and ET are analyzed.**

**Response:** We have added it as follows:

Forests change the surface energy, mass budgets, and water cycle by modifying the physical

properties of the land surface, such as albedo and roughness.

**Lines 27-32: The authors listed several papers describing the benefits of afforestation. More details would be helpful for readers to understand the impacts of afforestation from the process level.**

**Response:** We have included the details as follows:

Afforestation could increase carbon stocks in terrestrial ecosystems by absorbing atmospheric carbon dioxide through its biogeochemical effect (Jayakrishnan et al., 2023; Zhu et al., 2019; Gundersen et al., 2021). Meanwhile, afforestation changes the surface energy, mass budgets, and water cycle by modifying the physical properties of the land surface, such as albedo and roughness, and the partitioning between sensible and latent heat fluxes through biogeophysical progress as well (Bonan, 2008; Breil et al., 2021; Wang et al., 2023). Specifically, afforestation causes warming effects due to the decreased albedo and cooling effects due to increased evapotranspiration, which can partly offset or amplify the cooling effects due to taking up carbon from the atmosphere.

**Line 33: "Aggressively" is not a positive word.**

**Response:** We have deleted it.

**Line 36: Add the time constraint for global greening.**

**Response:** We have included it. "China's total forest cover has increased from 8.6 % in 1949 to 24.02 % in 2022 (Zhang and Song, 2006; Fu et al., 2023; Moore et al., 2016), resulting in a 42% contribution to the greening in China during 2000-2017 (Chen et al., 2019)."

**Line 44: Please refine this sentence: "trigger consequent effects on climate change, hydrological processes, carbon budget, ecosystem services".**

**Response:** We agree that this sentence is difficult to understand. We have refined it as follows:

Such large-scale afforestation in the future would modify the land cover conversions from non-forestland to forestland. These conversions could cause consequent effects on climate change (Wang et al., 2023), hydrological processes (Tian et al., 2022), carbon budget (Maneke-Fiegenbaum et al., 2021), ecosystem services (Wang and Li, 2022), etc.

References

Wang, H., Yue, C., & Luyssaert, S. (2023). Reconciling different approaches to quantifying land surface temperature impacts of afforestation using satellite observations. Biogeosciences, 20(1), 75-92.

Maneke-Fiegenbaum, F., Santos, S. H., Klemm, O., Yu, J. C., Chiang, P. N., & Lai, Y. J. (2021). Carbon dioxide fluxes of a young deciduous afforestation under the influence of seasonal precipitation patterns and frequent typhoon occurrence. Journal of Geophysical Research: Biogeosciences, 126(2), e2020JG005996.

Tian, L., Zhang, B., Chen, S., Wang, X., Ma, X., & Pan, B. (2022). Large-scale afforestation enhances precipitation by intensifying the atmospheric water cycle over the Chinese Loess Plateau. Journal of Geophysical Research: Atmospheres, 127(16), e2022JD036738.

Wang, Y., & Li, B. (2022). Dynamics arising from the impact of large-scale afforestation on ecosystem services. Land Degradation & Development, 33(16), 3186-3198.

**Line 45-46: Please provide the details for "sensitive to wetland reduction caused by afforestation" and "properties and intensities of these effects are highly dependent on the afforestation location and area. " I left confused about how the authors conclude.**

**Response:** Thank you for your suggestion. We replaced an example as follows:

It is crucial that the effects of afforestation are highly dependent on the afforestation location and area. For example, tropical afforestation leads to greater cooling effects than boreal afforestation (Arora and Montenegro, 2011). Therefore, it is urgent to arrange the national planned afforestation area to specific areas and project the possible land cover changes due to afforestation.

Reference

Arora, V. K., & Montenegro, A. (2011). Small temperature benefits provided by realistic afforestation efforts. Nature Geoscience, 4(8), 514-518.

**Line 49-55: The authors listed several papers and did not explain their findings on climate impact; in addition, please identify the deficiency of "employ idealistic and hypothetical**

**afforestation scenarios".**

**Response:** We have added the findings on climate impact as follows:

Odoulami et al. (2019) fully replaced the savanna areas (between 8°N and 12°N) with evergreen broadleaf trees over West Africa to investigate the climate effects of future afforestation. The obvious increase in the total annual precipitation was found over the afforested area. Similarly, Abiodun et al. (2013) employed random afforestation options to replace 25 %–100 % of the current land cover in Nigeria and found a local cooling effect induced by afforestation.

The deficiency of "employ idealistic and hypothetical afforestation scenarios" is that the afforestation scenarios were set by the authors themselves, and both the national afforestation plan and the future climate change constraint are neglected.

**Lines 63-71: Besides the dynamic downscaling, it would be beneficial to discuss the statistical downscaling. Moreover, dynamic vegetation studies for future projections in China are relevant to this topic, and the related studies should be mentioned in the literature review.**

**Response:** Following your suggestion, we will restructure it. We have added the dynamic vegetation and statistical downscaling studies in the revised manuscript as follows:

For statistical downscaling, the revision in the introduction is as follows:

However, the resolution of the raw GCM is much coarser (~100 km–300 km) to describe the fine land surface features at the regional scale (Varney, 2022; Turner et al., 2023; Song and Yan, 2022; Parsons, 2020). To overcome such shortage, downscaling techniques are widely used to translate GCM output to high-resolution data. Statistical downscaling involves the establishment of statistical relationships between local climate variables and large-scale atmospheric fields (Wilby and Wigley, 1997). However, it is not clear whether this historical statistical relationship is always stable in future periods. Statistical downscaling cannot ensure the physical consistency among meteorological variables. In contrast, the physically-based dynamic downscaling using a regional climate model (RCM) nested within a GCM could provide high-resolution climate simulations (Giorgi and Mearns, 1999; Mishra et al., 2014). The physical consistency is crucial to identify potential afforestation regions due to the multiple meteorological variables involved. Previous studies (Liu et al., 2020; Bowden et al., 2021) have employed the dynamical downscaling approach to quantify the climatological suitability for each nature vegetation type.

For statistical downscaling, the revision in the discussion is as follows:

Although the resolution of our dynamic downscaled simulation (25 km) is finer than raw GCMs (~100 km), it is difficult to meet the needs of afforestation planning in areas with complex topography. Convection-permitting climate modelling at the kilometre-scale has recently been developed to reproduce better mesoscale atmospheric processes (Prein et al., 2015; Lucas-Picher et al., 2021), and obviously improve the WRF simulation, especially precipitation (Knist et al., 2020). However, increasing the resolution of the simulation implies higher computational costs. In contrast, statistical downscaling methods are also known to obtain high-resolution climate data with few computational resources (Tang et al., 2016). The multi-model ensemble means from statistical downscaling CMIP6 can significantly reduce the biases compared to individual models (Gebrechorkos et al., 2019). Thus, some statistically downscaled CMIP6 datasets (Gebrechorkos et al., 2023; Lin et al., 2023; Thrasher et al., 2022), with a resolution of 0.1°-0.25° covering the global land, can be applied to explore the future global potential afforestation area in following work. However, it is noted that the statistically downscaling data may have a limitation, as the covariance among the variables may not align with physical laws.

For dynamic vegetation, the revision in the introduction is as follows:

In addition, process-based dynamic global vegetation models (DGVMs) are also useful tools to help quantify future afforestation scenarios (Krinner et al., 2005; Horvath et al., 2021). The DGVMs (i.e., LPJ-GUESS) have commonly been applied to explore the responses of potential natural vegetation distribution to climate change (Hickler et al., 2012; Verbruggen et al., 2021). The DGVMs driven by meteorological data generally consider complex biogeophysical, biogeochemical, and physiological progress, such as evapotranspiration, carbon–nitrogen interactions, photosynthesis, and so on (Cramer et al., 2001). Given that both model process parameters and future meteorological data from GCMs represent a large source of uncertainty in DGVMs, the double overlap can lead to great uncertainties (Jiang et al., 2012; Martens et al., 2021).

Reference

Hickler, T., Vohland, K., Feehan, J., Miller, P. A., Smith, B., Costa, L., et al. (2012). Projecting the future distribution of European potential natural vegetation zones with a generalized, tree species-based dynamic vegetation model. Global Ecology and Biogeography, 21(1), 50-63.

Krinner, G., Viovy, N., de Noblet-Ducoudré, N., Ogée, J., Polcher, J., Friedlingstein, P., et al. (2005). A dynamic global vegetation model for studies of the coupled atmosphere-biosphere system. Global Biogeochemical Cycles, 19(1).

Jiang, Y., Zhuang, Q., Schaphoff, S., Sitch, S., Sokolov, A., Kicklighter, D., & Melillo, J. (2012). Uncertainty analysis of vegetation distribution in the northern high latitudes during the 21st century with a dynamic vegetation model. Ecology and Evolution, 2(3), 593-614.

Martens, C., Hickler, T., Davis-Reddy, C., Engelbrecht, F., Higgins, S. I., Von Maltitz, G. P., et al. (2021). Large uncertainties in future biome changes in Africa call for flexible climate adaptation strategies. Global Change Biology, 27(2), 340-358.

Horvath, P., Tang, H., Halvorsen, R., Stordal, F., Tallaksen, L. M., Berntsen, T. K., & Bryn, A. (2021). Improving the representation of high-latitude vegetation distribution in dynamic global vegetation models. Biogeosciences, 18(1), 95-112.

Verbruggen, W., Schurgers, G., Horion, S., Ardö, J., Bernardino, P. N., Cappelaere, B., et al. (2021). Contrasting responses of woody and herbaceous vegetation to altered rainfall characteristics in the Sahel. Biogeosciences, 18(1), 77-93.

Giorgi, F., & Mearns, L. O. (1999). Introduction to special section: Regional climate modeling revisited. Journal of Geophysical Research, 104(D6), 6335–6352. https://doi.org/10.1029/98JD02072

Cramer, W., Bondeau, A., Woodward, F. I., Prentice, I. C., Betts, R. A., Brovkin, V., et al. (2001). Global response of terrestrial ecosystem structure and function to CO2 and climate change: results from six dynamic global vegetation models. Global change biology, 7(4), 357-373.

Prein, A. F., Langhans, W., Fosser, G., Ferrone, A., Ban, N., Goergen, K., et al. (2015). A review on regional convection-permitting climate modeling: Demonstrations, prospects, and challenges. Reviews of Geophysics, 53(2), 323-361.

Lucas-Picher, P., Argüeso, D., Brisson, E., Tramblay, Y., Berg, P., Lemonsu, A., et al. (2021). Convection-permitting modeling with regional climate models: Latest developments and next steps. Wiley Interdisciplinary Reviews: Climate Change, 12(6), e731.

Knist, S., Goergen, K., & Simmer, C. (2020). Evaluation and projected changes of precipitation statistics in convection-permitting WRF climate simulations over Central Europe. Climate Dynamics, 55(1-2), 325-341.

Tang, J., Niu, X., Wang, S., Gao, H., Wang, X., & Wu, J. (2016). Statistical downscaling and dynamical downscaling of regional climate in China: Present climate evaluations and future climate projections. Journal of Geophysical Research: Atmospheres, 121(5), 2110-2129.

Wilby, R. L., & Dawson, C. W. (2013). The statistical downscaling model: insights from one decade of application. International Journal of Climatology, 33(7), 1707-1719.

Gebrechorkos, S., Hülsmann, S., & Bernhofer, C. (2019). Regional climate projections for impact assessment studies in East Africa. Environmental Research Letters, 14(4), 044031.

Gebrechorkos, S., Leyland, J., Slater, L., Wortmann, M., Ashworth, P. J., Bennett, G. L., et al. (2023). A high-resolution daily global dataset of statistically downscaled CMIP6 models for climate impact analyses. Scientific Data, 10(1), 611.

Lin, H., Tang, J., Wang, S., Wang, S., & Dong, G. (2023). Deep learning downscaled high-resolution daily near surface meteorological datasets over East Asia. Scientific Data, 10(1), 890.

Thrasher, B., Wang, W., Michaelis, A., Melton, F., Lee, T., & Nemani, R. (2022). NASA global daily downscaled projections, CMIP6. Scientific Data, 9(1), 262.

**Line 67: Please talk about the uncertainties for GCMs.**

**Response:** Following your suggestion, we will restructure the discussion. We have added it as follows:

This study may have some limitations and uncertainties. Following the approach of existing studies (Ma et al., 2023; Qiu et al., 2022), we also utilize the bias-correction LBC in dynamical downscaling. However, the model uncertainty in the future climate projection is difficult to quantify because one GCM is used to nest into the WRF model. The projected result generally exhibits variations based on the choice of driving GCMs (Gao et al., 2022). This divergence can be attributed to the inherent configurations and physics parameterization of the GCMs, distinct radiative forcing scenarios, and varying equilibrium climate sensitivities found in CMIP6 models (Zuo et al., 2023; Bukovsky and Mearns, 2020). For instance, the high emissions scenario could lead to higher temperature and stronger precipitation in China (Yang et al., 2021). Consequently, the suitability of land for future forests may change accordingly. Exploring the impacts of different SSPs on the distribution of potential afforestation regions would be an intriguing avenue for future research. To address the concerns on model uncertainty, using WRF forced by multiple bias-correction CMIP6

model can explore the source of uncertainty, and the ensemble means for downscaled climate simulation would help to obtain a more robust projection. In addition, the different combinations of physics parameterization schemes in the WRF model also influence the simulation performance (Gbode et al., 2019). Selecting the optimal combination is beneficial for reducing underlying bias.

References

Ma, M., Tang, J., Ou, T., & Zhou, P. (2023). High-resolution climate projection over the Tibetan Plateau using WRF forced by bias-corrected CESM. Atmospheric Research, 286, 106670.

Qiu, Y., Feng, J., Yan, Z., Wang, J., & Li, Z. (2022). High-resolution dynamical downscaling for regional climate projection in Central Asia based on bias-corrected multiple GCMs. Climate Dynamics, 58(3-4), 777-791.

Gao, S., Zhu, S., & Yu, H. (2022). Dynamical downscaling of temperature extremes over China using the WRF model driven by different lateral boundary conditions. Atmospheric Research, 278, 106348.

Zuo, Z., Fung, J. C., Li, Z., Huang, Y., Wong, M. F., Lau, A. K., & Lu, X. (2023). Projection of future heatwaves in the Pearl River Delta through CMIP6-WRF dynamical downscaling. Journal of Applied Meteorology and Climatology, 62(9), 1297-1314.

Bukovsky, M. S., & Mearns, L. O. (2020). Regional climate change projections from NA-CORDEX and their relation to climate sensitivity. Climatic Change, 162(2), 645-665.

Yang, X., Zhou, B., Xu, Y., & Han, Z. (2021). CMIP6 evaluation and projection of temperature and precipitation over China. Advances in Atmospheric Sciences, 38, 817-830.

Gbode, I. E., Dudhia, J., Ogunjobi, K. O., & Ajayi, V. O. (2019). Sensitivity of different physics schemes in the WRF model during a West African monsoon regime. Theoretical and Applied Climatology, 136, 733-751.

**Lines 79-80: please add sequence numbers for three categories.**

**Response:** We have added it as follows:

This study used three categories of data: (1) ground meteorology measurements data, satellite-observed land use/cover data, (2) national planned afforestation area data, (3) climate modelling data from GCM, and ERA5 reanalysis data.

**Figure 2: The red text on a dark blue background is hard to read.**

**Response:** We changed it from red to white to make it easier to read.

[Figure]

| 1. Heilongjiang | 10. Jiangsu | 19. Anhui | 27. Yunnan |
| 2. Jilin | 11. Shannxi | 20. Hubei | 28. Fujian |
| 3. Liaoning | 12. Inner Mongolia | 21. Chongqing | 29. Guangdong |
| 4. Beijing | 13. Ningxia | 22. Sichuan | 30. Guangxi |
| 5. Tianjin | 14. Gansu | 23. Zhejiang | 31. Hainan |
| 6. Hebei | 15. Qinghai | 24. Jiangxi | 32. Taiwan |
| 7. Shanxi | 16. Xinjiang | 25. Hunan | 33. Hong Kong |
| 8. Shandong | 17. Tibet | 26. Sichuan | 34. Macau |
| 9. Henan | 18. Shanghai | | |

Figure 1: Model domain with topography. The black boundaries indicate each province in China.

**Lines 162-163: Did the authors test whether the model has reached the equilibrium state with only one year of spinning up?**

**Response:** The spin-up time of the WRF model is important to reach the physical equilibrium state and to avoid inhomogeneities. Its length is determined by the quality of initial conditions inputs. In this study, we use the ERA5 reanalysis data and the MPI–ESM1–2–HR model as the initial conditions. They generally reach the equilibrium state in a short time due to the physical consistency between the variables. Previous studies have demonstrated that 4- to 8-day for atmospheric variables and 1-year spin-up time for soil moisture and temperature is enough (Zhong et al., 2007; Katragkou et al., 2015). Therefore, the model can reach the equilibrium state in this study.

Reference

Gao, S., Huang, D., Du, N., Ren, C., & Yu, H. (2022). WRF ensemble dynamical downscaling of precipitation over China using different cumulus convective schemes. Atmospheric Research, 271, 106116.

Tang, J., Lu, Y., Wang, S., Guo, Z., Lu, Y., & Fang, J. (2023). Projection of hourly extreme precipitation using the WRF model over eastern China. Journal of Geophysical Research: Atmospheres, 128(1), e2022JD036448.

Zhong, Z., Yijia, H. U., Jinzhong, M. I. N., & Honglei, X. U. (2007). Numerical experiments on the spin-up time for seasonal-scale regional climate modeling. Journal of Meteorological Research, 21(4), 409–419.

Katragkou, E., García Díez, M., Vautard, R., Sobolowski, S. P., Zanis, P., Alexandri, G., Cardoso, A., Colette, A., Fernandez, J., Gobiet, A., Goergen, K., Karacostas, T., Knist, S., Mayer, S., Soares, P. M. M., Pytharoulis, I., Tegoulias, I., Tsikerdekis, A., & Jacob, D. (2015). Regional climate hindcast simulations within EURO-CORDEX: Evaluation of a WRF multi-physics ensemble. Geoscientific Model Development, 8(3), 603–618.

**Line 164: Delete the space between FUT_  and MPI.**

**Response:** We have deleted it.

**Line 212: Please change the unit mu into a standard international unit.**

**Response:** We have changed "1.825 billion mu" to "$121.67 \times 10^4$ km$^2$ ".

**Figure 3: in addition to the difference in FigS2. A pattern correlation and RMSE for AT, TP, and PE in Fig.3 would be beneficial.**

**Response:** We have added it to the supplementary material.

[Figure]

Figure 2: Comparison of observation, HIS_ERA and HIS_MPI based on RMSE. HIS_ERA and HIS_MPI indicate the WRF simulation driven by ERA5 reanalysis data and bias-corrected MPI–ESM1–2–HR model, respectively. The observation derives from the CN05.1 dataset.

[Figure]

Figure 3: Comparison of observation, HIS_ERA and HIS_MPI based on spatial correlation coefficient. HIS_ERA and HIS_MPI indicate the WRF simulation driven by ERA5 reanalysis data and bias-corrected MPI–ESM1–2–HR model, respectively. The observation derives from the CN05.1 dataset.

**Figure 4: Maybe I missed something, but adding texts to identify the difference among a, b, and c would be helpful. More info in the caption also would be beneficial for reader to understand this figure. One interesting finding from the figure is that the model tends to underestimate the TP in the high value (>1600 mm) category and overestimate the PE in the high value (>3; unit?) category.**

**Response:** We have added more information to the caption of Figure 4. " Figure 4: Scatterplots of the annual average biotemperature (AT), annual total precipitation (TP), and potential evapotranspiration ratio (PE) for each grid against the observation and HIS_ERA, observation and HIS_MPI, HIS_MPI, and HIS_ERA. HIS_ERA and HIS_MPI indicate the WRF simulation driven by ERA5 reanalysis data and the bias-corrected MPI–ESM1–2–HR model, respectively. The observation derives from the CN05.1 dataset. Evaluation indexes included the bias, mean absolute error (MAE), and spatial correlation coefficient (R). The black dotted line indicates a 1:1 line."

We find that the simulated TP exceeding 1600 mm in southern China is underestimated and the simulated PE exceeding 3 in northwest China is overestimated. The WRF model generally overestimates light rain, and underestimates heavy rain, especially extreme precipitation (Mugume et al, 2018). Therefore, it is necessary to further improve the simulation accuracy of the WRF model.

Reference

Mugume, I., Basalirwa, C., Waiswa, D., Nsabagwa, M., Ngailo, T. J., Reuder, J., & Semujju, M. (2018). A comparative analysis of the performance of COSMO and WRF models in quantitative rainfall prediction. International Journal of Marine and Environmental Sciences, 12(2), 130-138.

**Figure 5: Please add some values for change in the Fig. 5b.**

**Response:** The flow diagram was improved and specific values are included.

[Figure]

Figure 4: Projected spatial pattern of (a) potential vegetable types from HLZ model under the SSP2–4.5 scenario in the future periods (2041–2060) from the FUT_ MPI simulation, and (b) area changes across historical baseline (1995–2014) and future periods, where the calculations are based on FUT_ MPI simulation versus HIS_ MPI simulation.

**Figure 6: Some text overlaps with the map.**

**Response:** We have changed it.

[Figure]

Figure 5: Spatial distribution of (a) historical open space region for afforestation, (b) future potential forestation domain (PFD) from HLZ model considered as the forest suitable lands, (c) potential afforestation region constrained by climate change, (d) national planned afforestation areas in the individual provinces from the NFMP, (e) Chinese vegetation regionalization map.

---

## Author Response (AR2)

**Response to Reviewer 3 Comments**

**I have carefully read the comments from other reviewers as well as the authors' responses and the updated version of the manuscript. The manuscript has largely improved and is nearly ready for publication. I only have a few minor suggestions for further enhancement:**

**Lines 35-40: As noted in my first-round review, the authors have listed several ecological projects. It would be beneficial for the readers to see the advantages of these projects highlighted in existing studies. This part should be further enhanced.**

**Liu, Y., Ge, J., Guo, W., Cao, Y., Chen, C., Luo, X., Yang, L. and Wang, S., 2023. Revisiting biophysical impacts of greening on precipitation over the Loess Plateau of China using WRF with water vapor tracers. Geophysical Research Letters, 50(8), p.e2023GL102809.**

**Response:** Thank you for your suggestion. We have included the advantages of ecological projects as follows:

L37-40: These ecological engineering program programs have been beneficial for water conservation (Liu et al., 2023), mitigating climate warming (Yu et al., 2020), increasing terrestrial carbon sequestration (Shi and Han, 2014), reducing water erosion risk (Wang et al., 2021), and alleviating dust storm (Tan and Li, 2015).

**References**

Wang, H., Zhao, W., Li, C., and Pereira, P.: Vegetation greening partly offsets the water erosion risk in China from 1999 to 2018, Geoderma, 401, 115319, doi:10.1016/j.geoderma.2021.115319, 2021.

Liu, Y., Ge, J., Guo, W., Cao, Y., Chen, C., Luo, X., Yang, L., and Wang, S.: Revisiting biophysical impacts of greening on precipitation over the Loess Plateau of China using WRF with water vapor tracers, Geophys. Res. Lett., 50(8), e2023GL102809, doi:10.1029/2023GL102809, 2023.

Yu, L., Liu, Y., Liu, T., and Yan, F.: Impact of recent vegetation greening on temperature and precipitation over China, Agr. Forest. Meteorol., 295, 108197, doi:10.1016/j.agrformet.2020.108197, 2020.

Tan, M., and Li, X.: Does the Green Great Wall effectively decrease dust storm intensity in China? A study based on NOAA NDVI and weather station data, Land Use Policy, 43, 42-47, doi:10.1016/j.landusepol.2014.10.017, 2015.

Shi, S., and Han, P.: Estimating the soil carbon sequestration potential of China's Grain for Green Project, Global. Biogeochem. Cy., 28(11), 1279-1294, doi:10.1002/2014GB004924, 2014.

**Lines 55-60: I concur with reviewers 1 and 2 that a discussion on the temperature impacts for here is desired.**

**Response:** In the revised manuscript, the temperature impacts are discussed, as follows:

L402-411: Afforestation can provide temperature benefits (e.g., cooling the land surface) according to previous studies (Peng et al., 2014; Yu et al., 2020; Breil et al., 2024). However, the biophysical response of afforestation on temperature varies spatially. At a global scale, it is common sense that afforestation causes the warming effect in high-latitude regions due to the albedo-dominant radiation effect, while the cooling effect in low-latitude regions due to the evapotranspiration-dominant non-radiation effect (Bonan, 2008; Arora and Montenegro, 2011). Thus, afforestation-induced regional temperature changes depend on the net effects. Afforestation also can cause daytime cooling but nighttime warming (Yuan et al., 2022), and increase the surface temperature in winter, but decrease in other seasons (Ma et al., 2017). Differential responses in season and daily lead to more larger uncertainties in the net effects induced by afforestation. Therefore, a more realistic afforestation scenario is necessary to quantify the effects of afforestation on temperature under future climate change background and develop climate change mitigation policies.

**References**

Peng, S., Piao, S., Zeng, Z., Ciais, P., Zhou, L., Li, L. Z. X., Myneni, R. B., Yin, Y., and Zeng, H.: Afforestation in China cools local land surface temperature, P. Natl. Acad. Sci. USA, 111, 2915–2919, doi:10.1073/pnas.1315126111, 2014.

Yu, L., Liu, Y., Liu, T., and Yan, F.: Impact of recent vegetation greening on temperature and precipitation over China, Agr. Forest. Meteorol., 295, 108197, doi:10.1016/j.agrformet.2020.108197, 2020.

Arora, V. K., and Montenegro, A.: Small temperature benefits provided by realistic afforestation efforts, Nat. Geosci., 4(8), 514-518, doi:10.1038/ngeo1182, 2011.

Ma, W., Jia, G., and Zhang, A.: Multiple satellite‑based analysis reveals complex climate effects of

temperate forests and related energy budget, J. Geophys. Res.-Atmos., 122(7), 3806-3820, doi:10.1002/2016JD026278, 2017.

Yuan, G., Tang, W., Zuo, T., Li, E., Zhang, L., and Liu, Y.: Impacts of afforestation on land surface temperature in different regions of China, Agr. Forest. Meteorol., 318, 108901, doi:10.1016/j.agrformet.2022.108901, 2022.

Bonan, G. B.: Forests and climate change: forcings, feedbacks, and the climate benefits of forests, Science, 320(5882), 1444–480 1449, doi:10.1126/science.1155121, 2008.

Breil, M., Schneider, V. K. M., and Pinto, J. G.: The effect of forest cover changes on the regional climate conditions in Europe during the period 1986–2015, Biogeosciences, 21, 811–824, doi:10.5194/bg-21-811-2024, 2024.

**The analysis in this paper is based on the SSP 245 scenario, which the authors claim to be the most reliable. I have reservations about this assertion and believe that SSP 370 might be a more appropriate choice, considering its widespread use in large ensemble simulations such as LENS2. While the authors have provided some info on this topic, a more thorough discussion of the scenario impacts on the results and conclusions of this manuscript is necessary.**

**Liu, W., Wang, G., Yu, M., Chen, H., Jiang, Y., Yang, M. and Shi, Y., 2020. Projecting the future vegetation–climate system over East Asia and its RCP-dependence. Climate Dynamics, 55, pp.2725-2742.**

**Response:** We have included the discussions as follows:

L429-434: For instance, the high emission scenario could lead to higher temperature and stronger precipitation in China relative to middle emission (Yang et al., 2021). The obvious differences are found in the northern China. It implies that there are greater opportunities for afforestation in semi-arid areas. Thus, the suitability of future forest lands depends on emission scenarios (Liu et al., 2020; Elsen et al., 2022). Exploring the impacts of different SSPs on the distribution of potential afforestation regions would be an intriguing avenue for future research.

**References**

Liu, W., Wang, G., Yu, M., Chen, H., Jiang, Y., Yang, M., and Shi, Y.: Projecting the future

vegetation–climate system over East Asia and its RCP-dependence, Clim. Dyn, 55, 2725-2742, doi:10.1007/s00382-020-05411-2, 2020.

Yang, X., Zhou, B., Xu, Y., and Han, Z.: CMIP6 evaluation and projection of temperature and precipitation over China, Adv. Atmos. Sci., 38, 817–830, doi:10.1007/s00376-021-0351-4, 2021.

Elsen, P. R., Saxon, E. C., Simmons, B. A., Ward, M., Williams, B. A., Grantham, H. S., Kark, S., Levin, N., Perez-Hammerle, K. V., Reside, A. E., and Watson, J. E. M.: Accelerated shifts in terrestrial life zones under rapid climate change, Global. Change. Biol., 918–935, https://doi.org/10.1111/gcb.15962, 2022.

**I strongly recommend making all data openly accessible online to facilitate further research within the scientific community.**

**Response:** The future potential afforestation distribution data is available at https://zenodo.org/records/10900150.